# Neural sensitization improves encoding fidelity in the primate retina

Todd R. Appleby [1,2,3] & Michael B. Manookin [2,3]

An animal's motion through the environment can induce large and frequent fluctuations in light intensity on the retina. These fluctuations pose a major challenge to neural circuits tasked with encoding visual information, as they can cause cells to adapt and lose sensitivity. Here, we report that sensitization, a short-term plasticity mechanism, solves this difficult computational problem by maintaining neuronal sensitivity in the face of these fluctuations. The numerically dominant output pathway in the macaque monkey retina, the midget (parvocellular-projecting) pathway, undergoes sensitization under specific conditions, including simulated eye movements. Sensitization is present in the excitatory synaptic inputs from midget bipolar cells and is mediated by presynaptic disinhibition from a wide-field mechanism extending >0.5 mm along the retinal surface. Direct physiological recordings and a computational model indicate that sensitization in the midget pathway supports accurate sensory encoding and prevents a loss of responsiveness during dynamic visual processing.

[1] Graduate Program in Neuroscience, University of Washington, Seattle, WA 98195, USA. [2] Department of Ophthalmology, University of Washington, Seattle, WA 98195, USA. [3] Vision Science Center, University of Washington, Seattle, WA 98195, USA. Correspondence and requests for materials should be addressed to M.B.M. (email: manookin@uw.edu)

The fundamental constraints on sensory coding require that neural circuits adjust their outputs based on the statistical properties of their recent inputs[1–3]. Neurons respond to dynamic inputs using two distinct strategies—adaptation and sensitization. Adapting cells respond to strong stimulation by decreasing their sensitivity and this decrease in responsiveness can persist for several seconds after the stimulus intensity decreases[3–9]. Thus, adapting cells are relatively insensitive to weak stimuli occurring during these transition periods. Sensitizing cells show the opposite pattern—increasing their responsiveness at these transitions[10–12]. For this reason, adaptation and sensitization are commonly thought to constitute opposing and complementary forms of short-term neural plasticity[10,11].

This hypothesis requires that a sensitizing cell type have an adapting counterpart that encodes common information[10]. However, this constraint could potentially decrease the amount of information that can be encoded in an neural ensemble and increase the metabolic demands on a sensory tissue[3,13]. Nowhere is the need for metabolic and encoding efficiency more evident than in the macula of the primate retina where the tight packing of cells places space and metabolic resources at a premium. Alternatively, adaptation and sensitization might be signatures of fundamentally distinct neural coding strategies[14]. Further, these alternative hypotheses are not mutually exclusive—adapting and sensitizing cells could mirror each other in some species and neural pathways and not in others, depending on the particular coding and metabolic constraints in those systems[2,3,13,15]. However, given that neural sensitization was only recently discovered, relatively little is known about its roles in neural information processing.

To address this issue, we recorded from five types of output neurons in the macaque monkey retina—broad thorny (koniocellular-projecting), On and Off parasol (magnocellular-projecting), and On and Off midget (parvocellular-projecting) ganglion cells. These cells have well described roles in visual processing and no known functional counterparts. We studied how these cells responded to global fluctuations in contrast and other stimulus statistics. We report that whereas broad thorny and parasol cells strongly adapted, midget cells sensitized—increasing their responsiveness to certain types of visual stimulation, including simulated eye movements. Synaptic current recordings revealed that this increased sensitivity was present in the excitatory input from midget bipolar cells and was mediated by presynaptic disinhibition. The mechanism that increased sensitivity in midget cells originated in the receptive-field surround and showed a spatial extent in excess of 0.5 mm (>2.5 degrees). A computational model based on synaptic input recordings further indicated that this increase in sensitivity greatly enhanced the fidelity of encoding natural scenes. Moreover, the lack of an adapting counterpart to midget cells indicated that sensitizing circuits performed a distinct role in primate retina relative to that observed in other vertebrate neural systems[10–12,16].

## Results

### Midget ganglion cells exhibit contrast sensitization/facilitation.
The midget pathway of the primate retina is commonly believed to lack short-term plasticity mechanisms such as contrast gain control. This belief is based on reports that midget cells do not exhibit noticeable changes in responsiveness following transitions from high to low-contrast regimes[9,17]. The assay used to measure adaptation was a sinusoidally modulated drifting grating in which contrast was high for several seconds after which it transitioned to low contrast. Following the offset of high-contrast stimulation, midget cells did not exhibit a noticeable change in spiking relative to the period prior to the onset of high-contrast stimulation. This

was in stark contrast to the behavior observed in parasol ganglion cells. Parasol cells showed a strong and persistent depression in spiking following the offset of high contrast—behavior consistent with cells undergoing contrast adaptation[9,17].

The grating stimulus that did not elicit visible adaptation in midget cells was comprised of a spatial frequency tuned to the size of each cell's receptive-field center, which is narrower than many other retinal cell types. Thus, if plasticity in the midget pathway depended on mechanisms with broader spatial tuning, this assay would not have engaged these mechanisms.

To determine whether short-term plasticity in the midget pathway depends on the spatial properties of the stimulus, we repeated this assay while varying the spatial tuning of the gratings. Consistent with previous reports[9,17], midget cells did not exhibit a notable change in firing following the offset of a high contrast, high spatial frequency grating relative to the period that preceded high-contrast stimulation (Fig. 1a). To determine whether this lack of either adaptation or sensitization persisted across a range of stimulus conditions, we varied the spatial frequency content of the drifting gratings. Following the offset of low spatial frequency gratings, most midget cells showed an increase in spiking relative to the period preceding grating onset (Fig. 1b). This increase in spiking following high contrast is characteristic of the contrast sensitization observed in other vertebrate retinas[10–12]. The presence of sensitization at low spatial frequencies suggested that it depended on the ability to engage elements in the midget cell receptive field with broad spatial tuning relative to the midget bipolar cell.

To ensure that this increase in spiking was not an artifact of the phase of the grating at the offset of high contrast, we randomized the grating phase on each trial in several cells. Indeed, randomizing the grating phase did not produce significant changes at grating offset relative to cells recorded with a fixed phase (midget cell $p$-value, 0.53; $n = 12$ midget cells; parasol cell $p$-value, 0.3; $n = 14$ parasol cells; Wilcoxon rank sum test).

### Stimulus dependence of contrast sensitization in midget ganglion cells.
Our next goal was to determine how this putative wide-field component of the midget cell receptive field contributed to contrast coding. To accomplish this goal, we sought a more spatiotemporally precise assay of sensitivity following wide-field adaptation. Contrast tuning of parasol and midget cells was determined with spots centered on the receptive field. Responses were measured in isolation (unadapted condition) or 50–100 ms following the offset of an adapting stimulus (adapted condition). The adapting stimulus was a large, high-contrast spot modulated at 12–30 Hz (diameter, 730 μm). Presentations of the adapted and unadapted stimuli were interleaved to account for any potential variability in cellular responses over time.

Example spike responses to this stimulus paradigm are shown in Fig. 2. Parasol cells increased their spike rate at the onset of the adapting stimulus and the spike rate quickly decreased to a steady-state rate by ~0.25 s. Test flashes presented after the offset of the adapting stimulus evoked fewer spikes relative to the unadapted control (Fig. 2a), resulting in a decrease in gain, defined as the steepest slope of the contrast-response function[9,17,18]. Both of these patterns—a transient increase in spike rate following the transition to high contrast and a decrease in spiking after the transition from high contrast—are characteristic of cells undergoing contrast adaptation[4,6,19]. This result confirms previous reports that parasol cells readily adapt to changes in contrast (Fig. 2g)[9,17,18].

Midget cells showed a different pattern—instead of reducing sensitivity as it did for parasol cells, the adapting stimulus significantly increased sensitivity in midget cells at low contrast

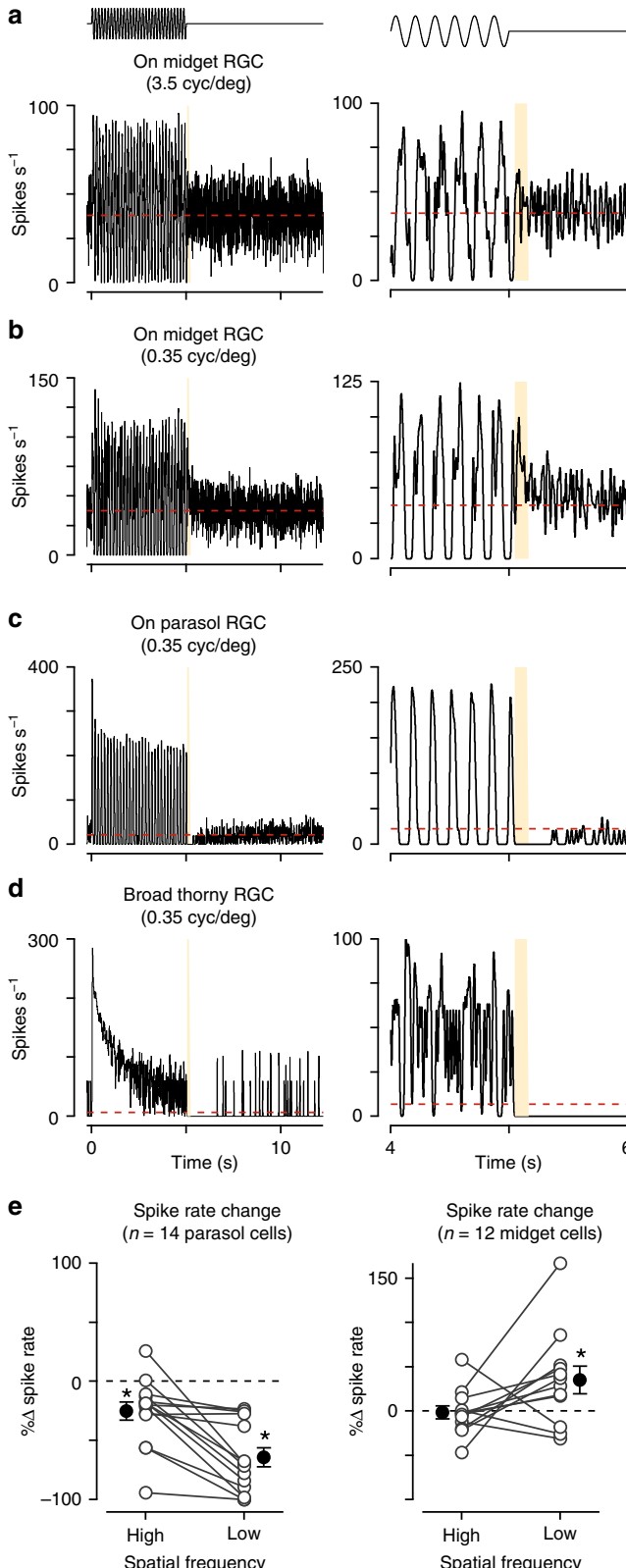

**Fig. 1** Midget and parasol ganglion cells exhibit opposing forms of plasticity. **a** Spike rate in an On midget ganglion cell to a high spatial frequency grating presented for 5 s (temporal frequency, 6 Hz; spatial frequency, 3.5 cycles degree$^{-1}$). The spike rate immediately after grating offset showed little change relative to the period prior to grating onset. Right: Zoom of transition period. **b** Spike responses from the same cell as in **a** to a low spatial frequency grating (0.35 cycles degree$^{-1}$). Spiking showed a transient increase following the offset of high contrast, consistent with contrast sensitization. **c** Spike rate in an On parasol ganglion cell to a low spatial frequency drifting grating (0.35 cycles degree$^{-1}$). After the offset of high contrast, the spike rate declined below the level prior to grating onset (red dashed line). **d** Same as **a** in a broad thorny (On-Off type) ganglion cell. **e** Change in spike rate for the period directly after grating offset relative to period prior to grating onset in parasol (left) and midget ganglion cells (right). Spiking in parasol cells was significantly reduced for both spatial frequencies ($p < 6.0 \times 10^{-3}$; $n = 14$ cells) and significantly increased in midget cells for the low spatial frequency grating ($p = 3.4 \times 10^{-2}$; $n = 12$ cells). Statistical significance calculated using the Wilcoxon signed rank test

test), indicating that sensitization improved contrast sensitivity in both the On and Off midget pathways.

We next tested whether contrast sensitization varied with the size of the test flash by repeating the adaptation experiment with wide-field test flashes to measure the contrast tuning of midget cells (diameter, 730 μm). Indeed, the adapting stimulus produced a significant increase in sensitivity relative to the unadapted condition at low contrast (Fig. 2i). These data indicated that sensitization persisted for wide-field stimulation. We further tested the time course of the sensitization effect and found that it could be engaged with relatively brief periods of stimulation (0.25 s) and persisted for ~0.4 s following the offset of high-contrast stimulation (Fig. 3).

Collectively, the experiments described thus far indicated that mechanisms within the midget cell receptive-field surround strongly contributed to contrast sensitization. To determine whether the observed plasticity mechanisms persisted for surround stimulation alone, we repeated the adapting stimulus paradigm while restricting both the adapting stimulus and the test probe to the receptive-field surround. Just as with full-field adaptation, surround adaptation significantly improved sensitivity at low contrast (Fig. 4). These results indicated that surround stimulation alone was sufficient to produce sensitization in midget cells. These data also indicated that the surround mechanism responsible for sensitization in the midget cells must be much larger than 160 μm in order to exert the observed effects on spiking in midget cells.

To estimate the spatial extent of the surround mechanism mediating contrast sensitization, we repeated the surround adaptation experiment using mask diameters of 160–640 μm in the same cell. Surround adaptation increased the spike output of midget cells to flashed annuli for mask diameters ≤480 μm (Fig. 4e). This increase in evoked spiking resulted in significant increases in sensitivity for a range of mask diameters (Fig. 4f). These data indicate that this mechanism must have a spatial extent well in excess of 480 μm in order to exert the observed effects on spiking in midget cells. This means that the sensitization mechanism we measured operates over spatial scales that are >10 times the size of the classical receptive-field center—the spatial extent of this effect narrows the candidate mechanisms to either horizontal cells in the outer retina or wide-field amacrine cells in the inner retina. Below, we present evidence that wide-field amacrine cells mediate contrast sensitization in the midget pathway.

(Fig. 2h). To determine whether these effects differed between On and Off midget cells, we separately analyzed the sensitivity metrics for On and Off cells for this stimulus paradigm. Following the adapting stimulus, contrast sensitivity significantly increased relative to the unadapted control in both On and Off midget cells at contrasts ≤0.25 ($p < 0.05$; Wilcoxon signed rank

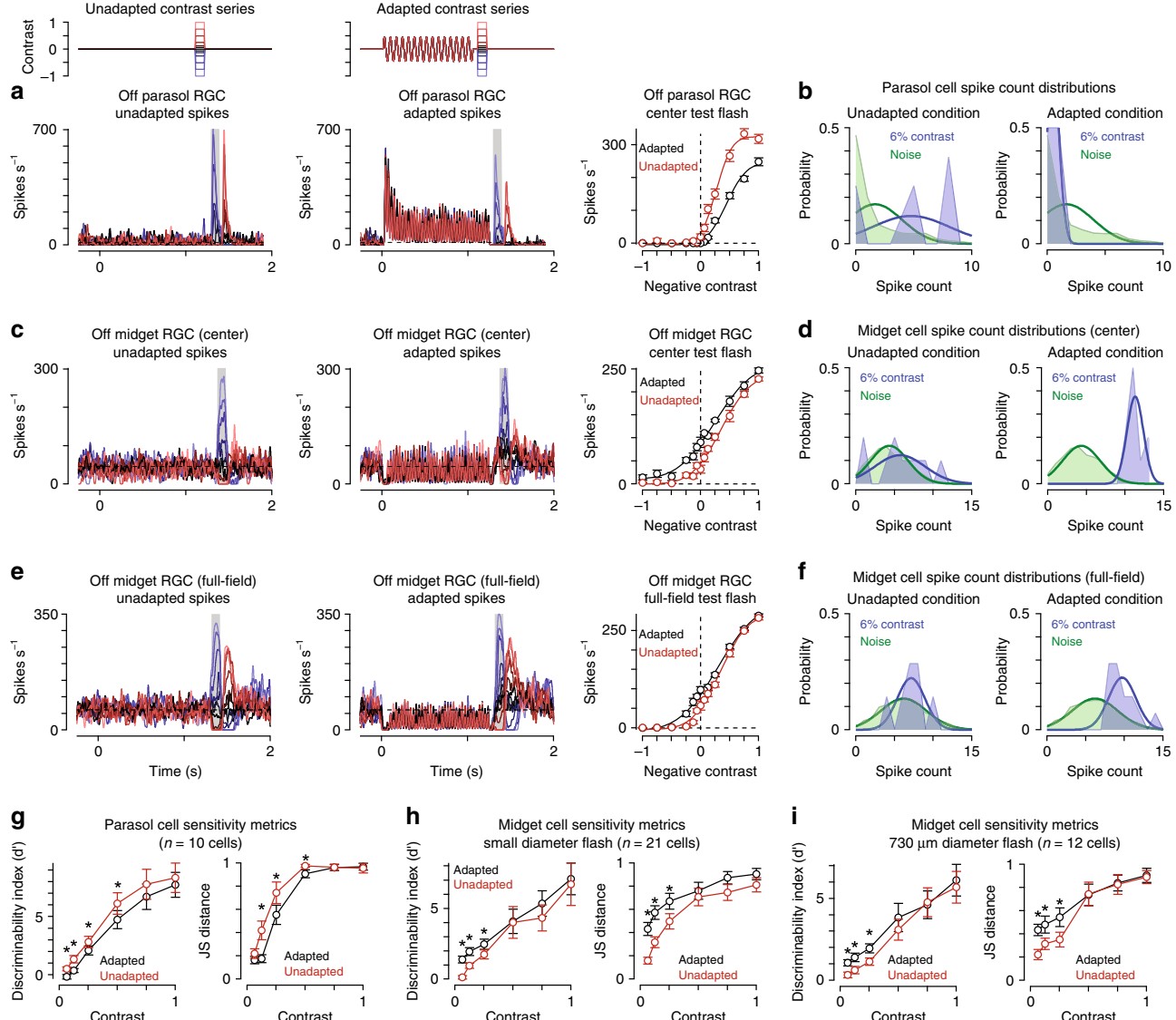

**Fig. 2** Midget ganglion cells display contrast sensitization. **a** Spike responses from an Off parasol ganglion cell to a series of spots centered over the receptive field. Spots were either presented alone (left) or 50 ms following the offset of an adapting stimulus (middle). Shaded regions indicate sampling windows. Right: Average spike rate across the shaded regions. The wide-field adaptation evoked a decrease in the slope (gain) of the contrast-response curve (black) relative to the unadapted control condition (red). **b** Spike count distributions for the Off parasol cell in (**a**) to a −6% contrast flash and the noise condition in which a flash was not presented (i.e., 0% contrast). The adapting stimulus decreased spiking at low contrast and shifted the low-contrast distribution toward the noise distribution (right) relative to the unadapted condition (left). **c** Same as (**a**) for an Off midget ganglion cell. Right: Average spike rate across the shaded regions. The wide-field adaptation evoked a leftward shift in the contrast-response curve (black) relative to the unadapted control condition (red). **d** Same as (**b**) for the Off midget cell in (**c**). The adapting stimulus increased spike counts at low contrast, increasing the separation between the low-contrast and noise distributions. **e** Spike responses of an Off midget cell to wide-field test flashes in the absence (left) or presence (middle) of the adapting stimulus. **f** Same as (**d**) for the Off midget cell in (**e**). **g** Discriminability index (left) and Jensen-Shannon distance (right) for preferred-contrast responses relative to background noise in parasol ganglion cells ($n = 10$). **h** Sensitivity indices in 21 midget ganglion cells for small diameter test flashes. Discriminability index and Jensen-Shannon distance increased significantly at low contrast (≤25%) following the adapting stimulus ($p < 0.01$). **i** Sensitivity indices in 12 midget cells for wide-field test flashes. The adapting stimulus produced a significant increase in the Jensen-Shannon distance at low contrast (6%; $p = 1.5 \times 10^{-3}$). Circles and bars indicate mean ± SEM. Statistical significance calculated using the Wilcoxon signed rank test

**Temporally uncorrelated stimuli reveal interactions between adaptive and sensitizing mechanisms.** To measure the effects of spatially localized changes in stimulus variance, we presented a randomly flickering spot over the receptive field (Fig. 5, top left). Cellular responses were modeled using the linear–nonlinear (LN) paradigm. Separate nonlinearities were calculated for the high-contrast condition, for the low-contrast region immediately following the transition from high to low contrast (low early; 100–600 ms following the transition), and for the more sustained period of low-contrast stimulation (low late). Changes in gain (the steepest slope of the nonlinearity) and horizontal offset were calculated by scaling the low contrast nonlinearities to match the nonlinearity at high contrast (see Methods section)[18].

Parasol cells showed a reduction in gain during the transition to low contrast relative to the later period of low contrast (Fig. 5a, h). Figure 5b illustrates the change in gain as a function of time for an

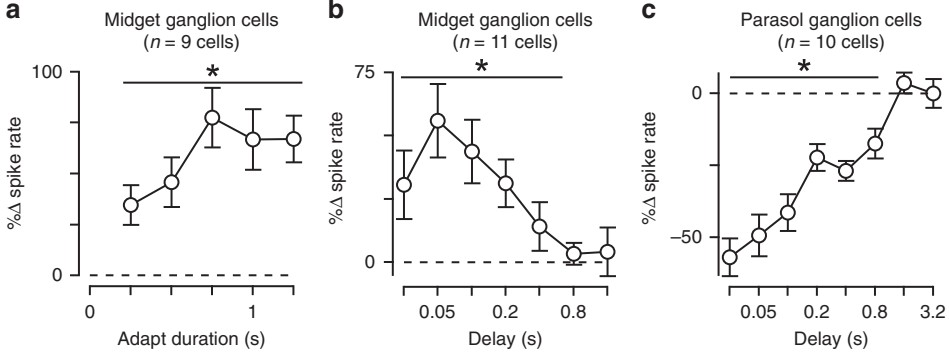

**Fig. 3** Time course of contrast sensitization and adaptation. **a** Change in spike rate for the adapted condition relative to unadapted control for adaptation periods (contrast, ±0.25–0.5; delay 0.05 s). Adaptation period was varied between 0.25 and 1.25 s (x-axis). The adapting stimulus produced a significant increase in spiking for each of the durations tested ($p < 4.0 \times 10^{-3}$; $n = 9$ cells). **b** Duration of contrast sensitization in midget ganglion cells. Test flashes (contrast, ±0.25–0.5) were presented at different delays (x-axis) following the offset of an adapting stimulus. Percent change in spike rate for the adapted condition relative to the unadapted condition is shown on the y-axis. Increase in spiking was statistically significant for delays ≤0.4 s ($p < 7.0 \times 10^{-3}$; $n = 11$ cells). **c** Same as (**b**) for parasol ganglion cells. The adapting stimulus significantly reduced spiking at delays ≤0.8 s ($p < 7.0 \times 10^{-3}$; $n = 10$ cells). Error bars indicate mean ± SEM. Statistical analyses were paired and significance was calculated using the Wilcoxon signed rank test

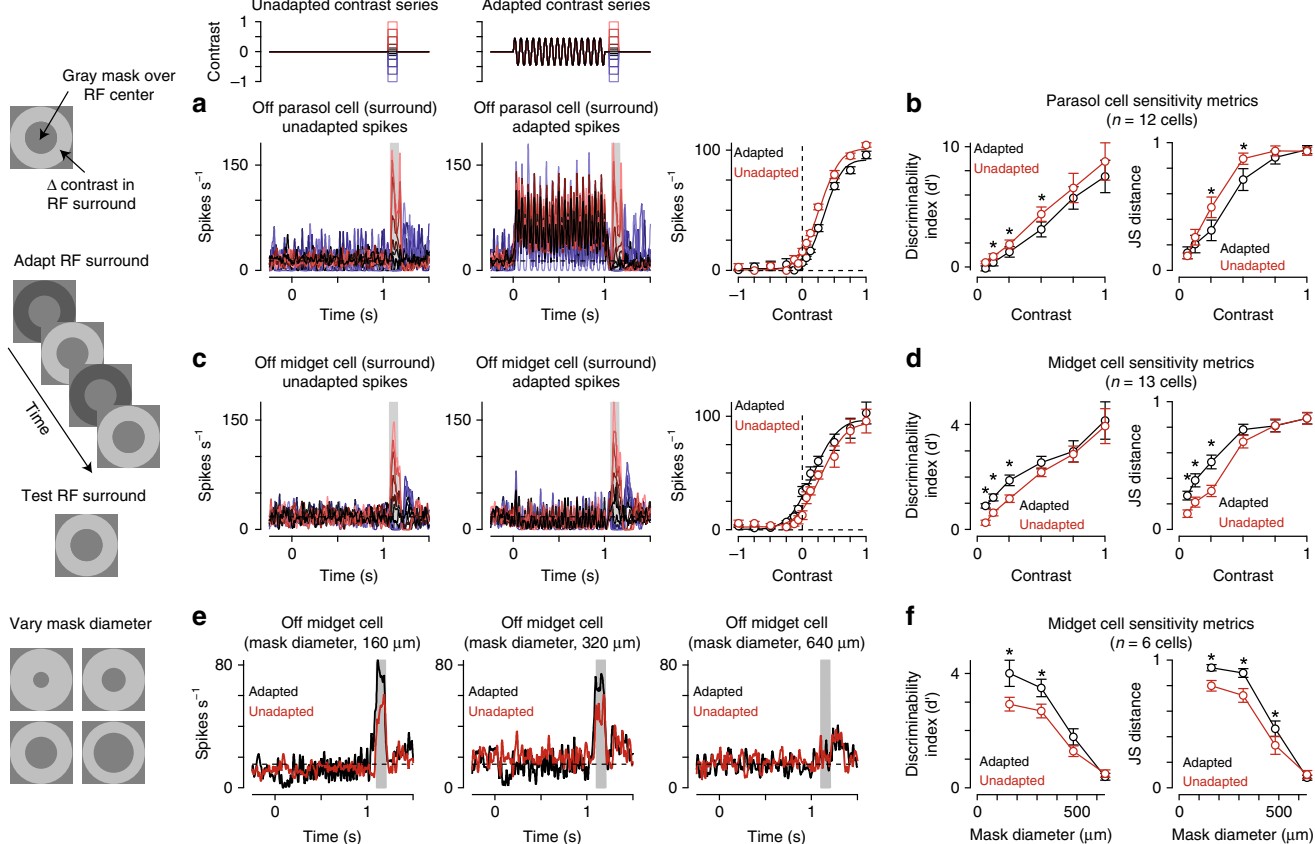

**Fig. 4** Sensitization arises in the receptive-field surround. **a** Spike responses from an Off parasol ganglion cell to a series of stimuli presented in the receptive-field surround. Annuli were presented in isolation (left) or following an adapting stimulus that was also presented in the receptive-field surround (middle). The adapting stimulus evoked a decrease in spiking at positive contrasts relative to the unadapted condition (red). **b** Discriminability index (d′; left) and Jensen-Shannon distance (right) for contrast responses relative to background noise in parasol ganglion cells ($n = 12$). Statistically significant change in sensitivity indicated with asterisk. **c** Responses from an Off midget ganglion cell to the surround adaptation stimulus paradigm. Consistent with weak contrast sensitization, the adapting stimulus elicited a slight leftward shift in the contrast-response function relative to the control condition (right). **d** Sensitivity indices in 12 midget ganglion cells for surround test flashes. **e** Sensitization varies with the degree of surround stimulation. Spike responses from an Off midget cell to the surround adaptation stimulus for three mask diameters. For some mask diameters, surround adaptation (black) produced larger spike responses during the surround flash relative to the unadapted condition (red). This increase in responsiveness was not present for the largest mask diameter (640 μm, right). **f** Sensitivity metrics for four mask diameters in midget ganglion cells ($n = 6$ cells). Surround adaptation produced significant larger d′ values for mask diameters ≤320 μm and Jensen-Shannon distance values for diameters ≤480 μm ($p < 0.05$; Wilcoxon signed rank test). Error bars indicate mean ± SEM

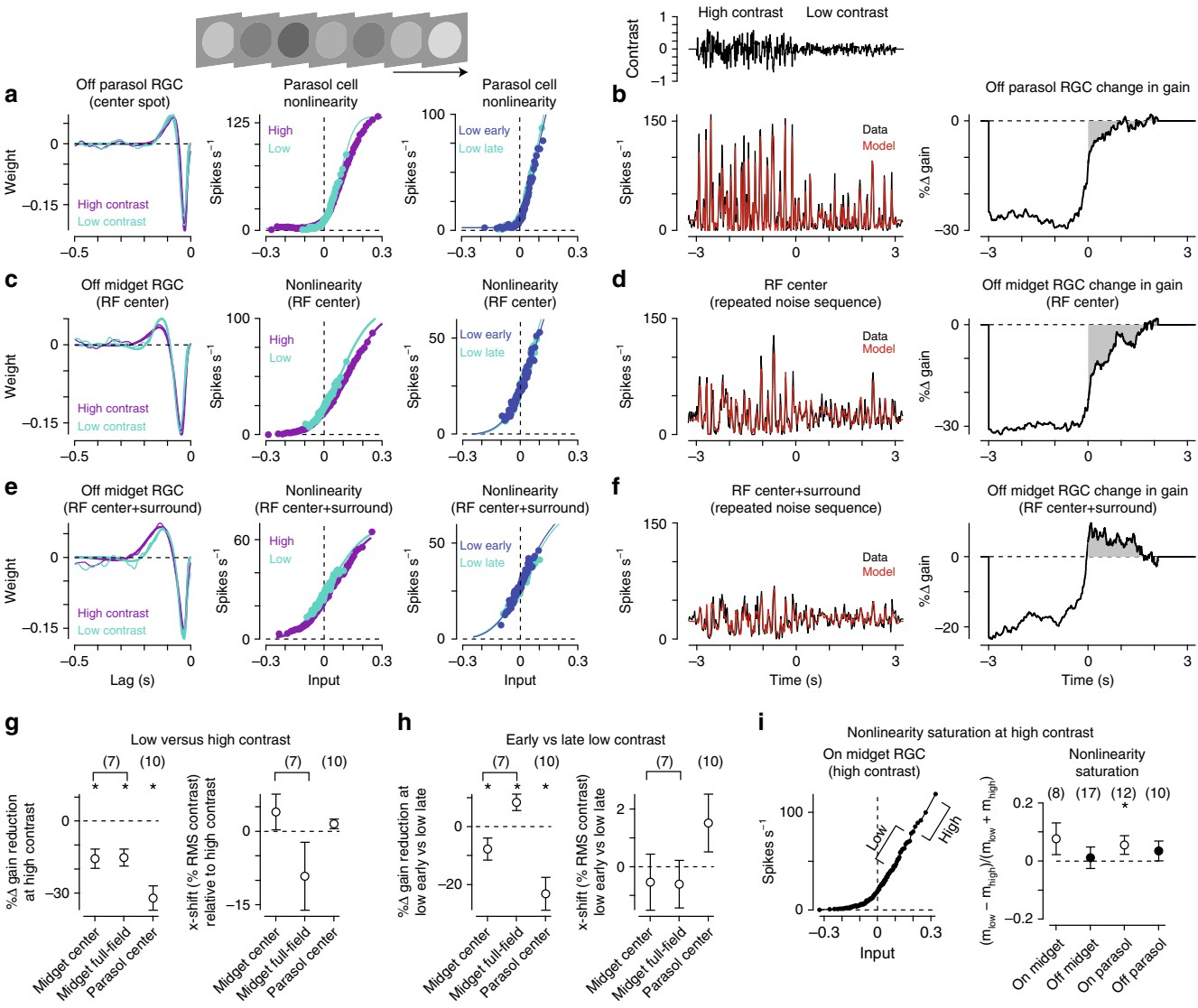

**Fig. 5** Changes in stimulus variance evoke changes in the input–output properties of midget cells. **a** Temporal filters (left) and input–output nonlinearities (middle) in an Off parasol cell for the high and low-contrast periods. Separate nonlinearities were also calculated for the low-contrast region directly following the transition from high-to-low contrast (low early) and for the sustained low-contrast region (low late; right). **b** Average spike rate (black) and linear–nonlinear model prediction (red) for the repeated contrast trajectory (left). Data and model showed high correspondence (high contrast $r^2$, 0.87 ± 0.02; low contrast $r^2$, 0.73 ± 0.04; $n = 10$ cells). Right, average gain as a function of time. Values are shown relative to the sustained low-contrast period. Shaded regions indicate the period of low contrast. **c** Same as **a** for an Off midget ganglion cell to a small spot (diameter, 80 μm) presented over the receptive-field center. **d** Left: Spike rate (black) and model prediction (red) for repeated contrast trajectory (high contrast $r^2$, 0.91 ± 0.02; low contrast $r^2$, 0.71 ± 0.07; $n = 7$ cells). The cell's gain was reduced following the shift to low contrast and recovered quickly to the level of sustained gain. **e** Linear–nonlinear model for the cell in (**c**) to a large spot (diameter, 730 μm). **f** Responses of the cell in (**b**) for the large diameter spot (left). Following the transition to low contrast, gain quickly increased, exceeding the sustained level for ~2 s (right). **g** Left: Reduction in gain at high contrast relative to the low-contrast condition. Gain was significantly reduced at high contrast for both midget and parasol cells ($n = 7$ midget cells; $n = 10$ parasol cells; $p < 0.05$). Right: Horizontal shift along the $x$-axis for the low contrast nonlinearity relative to high contrast. Wide-field stimulation in midget cells evoked a shift of ~10% RMS contrast for the low-contrast condition indicating that lower contrasts were required to evoke the same spike rate as the high contrast condition. **h** Change in gain (left) and horizontal shift (right) for the early versus late low-contrast periods. **i** Nonlinearity saturation was measured for high contrast Gaussian noise (RMS contrast, 0.3) in midget and parasol ganglion cells. Saturation was quantified by comparing the slopes for the high and low-variance regions of the nonlinearity (left). The slope differences were near zero, indicating a lack of strong saturation in both midget and parasol ganglion cells (right); only On parasol cells showed saturation values that were significantly greater than zero ($p = 4.6 \times 10^{-2}$; all other $p > 0.1$). Error bars indicate mean ± SEM. Statistical significance for paired values determined using Wilcoxon signed rank test and unpaired values with the Wilcoxon signed rank test

example Off parasol cell. Following the transition to low contrast, the cell's gain increased to a steady-state value over the course of ~1 s. This result was again consistent with strong contrast adaptation in parasol cells.

We measured responses from midget cells to the same stimulus protocol—the time-varying contrast trajectory was presented within a small spot over the receptive field (diameter, 40–80 μm). As with the parasol cells, midget cells showed a reduction in gain that slowly recovered following the transition to low contrast (Fig. 5c, d). This pattern of gain changes was consistent with weak contrast adaptation in midget cells to continuous stimulation that was localized to the cells' receptive-field center and near

surround. This result was surprising in the context of the other experiments presented thus far indicating that midget cells sensitized to stimulation of the receptive-field center + surround or of the surround alone. The differing results of these experiments and the noise experiments could arise from differences in the temporal frequency content of the stimuli— the stimuli eliciting sensitization were restricted to higher temporal frequencies (12–30 Hz) while the noise stimuli were temporally broad-band, equally sampling frequencies up to the Nyquist limit of the display (30 Hz). Another difference between these stimuli was their distinct spatial properties—the stimuli eliciting sensitization strongly modulated the receptive-field surround whereas the noise stimuli were primarily localized to the receptive-field center.

To distinguish between these possibilities, we repeated the noise experiments in the same cells using large diameter spots that stimulated both the receptive-field center and surround (diameter, 730 μm), and this stimulus elicited a distinct response pattern in midget cells. Following the transition to low contrast, gain rapidly increased after which it decreased gradually to the steady-state level (Fig. 5f). This increase in gain following the transition to low contrast was evidence that contrast sensitization depended on the spatial properties of the adapting stimulus and could be elicited with temporally white stimuli[10].

These results demonstrated that the midget cell receptive field contained two competing mechanisms contributing to short-term plasticity on different spatial scales—a center mechanism that weakly adapted to fluctuations in contrast and a surround mechanism that produced sensitization (i.e., prevented adaptation). We further tested how interactions between these narrow- and wide-field mechanisms contribute to visual coding in midget cells below. First, however, we tested whether midget cell responses saturated at high contrast similar to sensitizing cells in other species[10].

In salamander retina, sensitizing and adapting ganglion cells showed distinct sensitivity ranges. During high-contrast stimulation adapting cells reduced their gain, allowing them to avoid saturation and effectively encode contrast. However, their gain remained low for a time upon the shift to low contrast while the cells reestimated the stimulus variance[4,8,20]. Sensitizing cells showed the opposite pattern—their gain remained high during transitions to low contrast, allowing them to effectively encode contrast during these shifts from high-to-low variance conditions. The downside of this behavior is that the gain of sensitizing cells was high during periods of high contrast, which caused their responses to saturate. Based on these observations, it was proposed that adapting and sensitizing cells function in a concerted manner with adapting and sensitizing cells encoding in high and low-contrast regimes, respectively[10]. Thus, if the same pattern holds in primate retina, one would expect midget ganglion cells to saturate during high-contrast stimulation and one would likewise expect to find an adapting cell type that encodes common visual information to midget cells (i.e., red-green color, achromatic).

To determine whether midget cells saturated during high contrast, we recorded responses of these cells to a high-contrast Gaussian temporal flicker stimulus (s.d., 0.3). We quantified the degree of response saturation by calculating the slope of the nonlinearity for the high-variance region ($m_{high}$; >2 standard deviations) relative to the low-variance region ($m_{low}$; 1–2 standard deviations) of the nonlinearity.

$$\text{saturation index} = \frac{m_{low} - m_{high}}{m_{low} + m_{high}} \qquad (1)$$

Values near zero would occur for nonlinearities with little or no saturation while values near one occur for strongly sigmoidal

nonlinearities that saturated at high contrast. Consistent with a lack of saturation, index values were near zero for both On and Off midget cells (Fig. 5g). Thus, unlike sensitizing cells in other species, midget cells did not show saturation at high contrast[10], indicating that an adapting counterpart was not required to offset saturation in midget cells. This further suggested that sensitization performs a distinct role in midget cells relative to that posited for other species.

**Sensitization enhances chromatic processing in midget cells.** Midget ganglion cells in the central retina exhibit strong chromatic opponency which is formed from differential input from long-wavelength cones (L cones) and middle-wavelength cones (M cones) to the receptive-field center and surround[21–23]. To determine whether sensitization affected chromatic processing, we measured contrast responses in midget cells with purely chromatic (isoluminant) test flashes following the adapting stimulus.

Isoluminant (equiluminant) stimuli are commonly employed to study color mechanisms in isolation. We measured contrast-responses to purely chromatic (L − M; isoluminant) flashes (duration, 0.1 s) in the presence or absence of an achromatic adapting stimulus, as above. This stimulus was specifically designed to modulate chromatic mechanisms that differentiate L- and M-cone inputs (L − M; isoluminant) while silencing achromatic mechanisms that sum inputs from the L- and M-cone pathways (L + M; isochromatic).

As with the achromatic stimuli, the adapting stimulus significantly improved sensitivity for low-contrast test flashes (Fig. 6c), indicating that contrast sensitization enhanced both achromatic and chromatic processing in midget cells. While chromatic processing was affected by sensitization, the observation that an achromatic adapting stimulus was sufficient to evoke sensitization demonstrated that chromatic stimuli were not necessary to elicit the phenomenon. These data did not, however, rule out contributions from purely chromatic mechanisms to contrast sensitization.

To determine whether such a chromatic mechanism contributed to the observed contrast sensitization, we repeated the chromatic sensitivity tests following a chromatic adapting stimulus. However, the chromatic adapting stimulus did not produce a significant increase in chromatic contrast sensitivity at any contrast (Fig. 6f). We interpret this result as evidence that contrast sensitization arose from an achromatic mechanism in the midget cell receptive field. Moreover, given the role of horizontal cells in forming the L-versus-M opponent receptive-field surround, these data excluded horizontal cells as the source of sensitization in the midget pathway[21].

**Sensitization is present in excitatory synaptic input from midget bipolar cells.** The experiments above found contrast sensitization in the spike output of midget ganglion cells. Our next goal was to understand the circuit mechanisms mediating sensitization. To accomplish this goal, we measured the direct excitatory and inhibitory synaptic inputs to midget ganglion cells with whole-cell, voltage-clamp recordings (see Methods). Excitatory currents were isolated by holding a cell's membrane voltage at the reversal potential for inhibition (−70 mV), and likewise, inhibitory currents were recorded at the excitatory reversal potential (0 mV). An increase in excitatory input to a cell was indicated by a more negative (inward) current relative to the leak current. Indeed, the adapting stimulus evoked larger inward excitatory currents relative to the unadapted control at all contrasts tested (Fig. 7a). Plotting excitatory charge as a function of contrast revealed a similar pattern to that observed in the spike

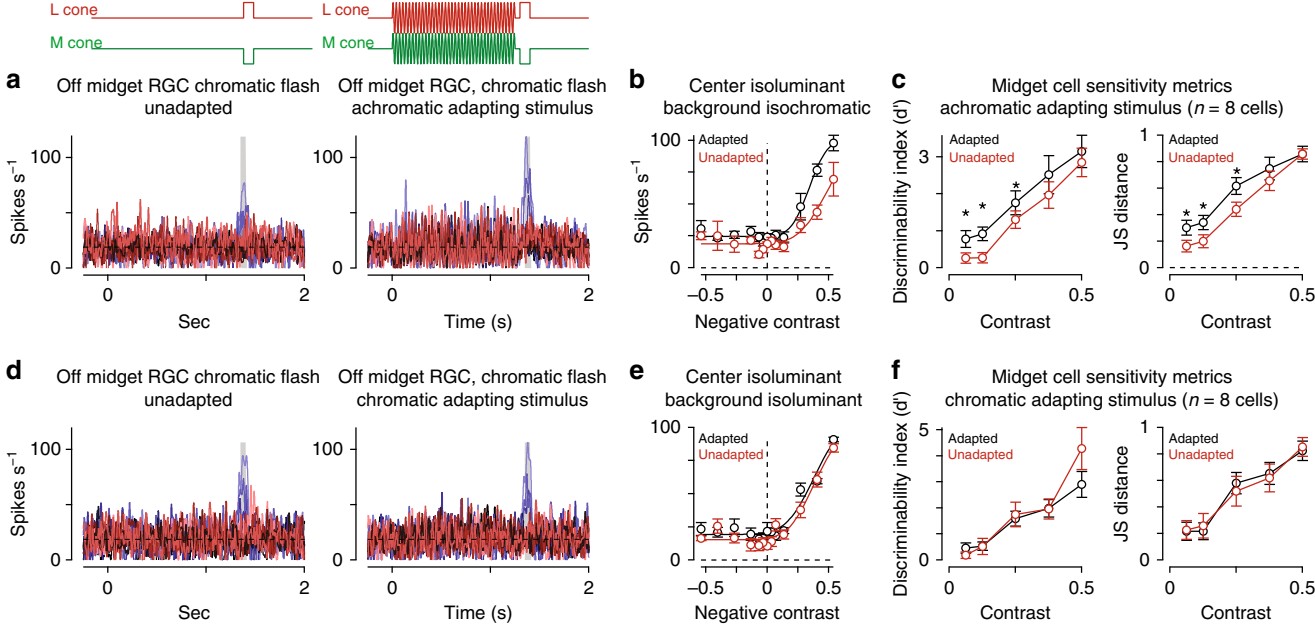

**Fig. 6** Sensitization arises from an achromatic mechanism. **a** Spike responses from an Off midget ganglion cell to a chromatic (isoluminant) contrast series. Spots were either presented alone (left) or 50 ms following the offset of an achromatic adapting stimulus (right). Shaded regions indicate sampling windows. **b** Average spike rate across the shaded regions indicated in (**a**). Achromatic adaptation evoked a leftward shift in the contrast-response curve (black) relative to the unadapted control condition (red) for the chromatic test flash. **c** Sensitivity metrics for the achromatic adapting stimulus followed by a chromatic contrast series in eight midget ganglion cells. The adapting stimulus improved chromatic sensitivity at low contrast (contrast, ≤25; $p < 0.05$; Wilcoxon signed rank test). **d** Spike responses for the cell in (**a**) to a chromatic adapting stimulus. **e** Average spike rate during the chromatic test flashes. The chromatic adapting stimulus did not evoke a large change in the contrast-response curve relative to control. **f** Sensitivity metrics for the chromatic adaptation experiment. Changes in sensitivity were not significantly different relative to the unadapted control at any contrast ($n = 8$ cells; $p > 0.1$; Wilcoxon signed rank test). Circles and bars indicate mean ± SEM

recordings—the adapting stimulus evoked a leftward shift in the contrast-response curve relative to the unadapted control (Fig. 7b). These results indicated that contrast sensitization was present in the excitatory synaptic input from midget bipolar cells to midget ganglion cells.

We also tested for the presence of sensitization in the inhibitory synaptic inputs to midget cells. Unlike the pattern observed in spiking and excitatory currents, the adapting stimulus did not elicit significant shifts in the inhibitory contrast-response functions relative to control (Fig. 7c). These data showed that contrast sensitization arose at or prior to the level of glutamate release from midget bipolar cells. This finding was consistent with the circuit model for contrast sensitization in bipolar cells in the retinas of fish, salamander, mice, and rabbits[10–12]. This model posited a mechanism in which a strongly adapting amacrine cell drove sensitization by a mechanism of presynaptic inhibition at the bipolar cell terminal[11]. During the adapting stimulus, the amacrine cell adapted such that, following stimulus offset, the cell decreased release of inhibitory neurotransmitter to the bipolar cell synaptic terminal relative to the tonic level. This presynaptic disinhibition, in turn, depolarized the bipolar cell synaptic terminal, allowing the cell to utilize its full dynamic range in signaling via glutamate release to postsynaptic ganglion cells (Fig. 8).

Cleanly measuring the effects of presynaptic inhibition on circuit function has proven exceedingly difficult as use of inhibitory receptor antagonists typically cause many off-target effects that make data interpretation highly tenuous[24]. Indeed, adding inhibitory antagonists in primate retina evoked significant increases in tonic glutamate release from bipolar cells and changed the contrast polarity of On parasol cells[25]. Nonetheless, our spike and whole-cell recordings strongly supported the proposed model in which contrast sensitization arose from

disinhibition at the presynaptic bipolar cell terminal[11]. First, the lack of sensitization to a purely chromatic (isoluminant) adapting stimulus indicated that sensitization did not arise in the outer retina at the level of horizontal cell feedback (Fig. 6). Further, horizontal cells are unlikely to contribute significantly to contrast sensitization as work in other species indicates that these cells do not exhibit contrast adaptation[4,26,27]. Second, the effect of presynaptic disinhibition was seen in our excitatory current recordings (Fig. 7e). In one of our stimulus conditions the test flash contrast was zero such that the stimulus intensity returned to the average background intensity at the offset of the adapting stimulus. Although this stimulus lacked a change in contrast following the adapting stimulus, we observed a significant increase in excitatory synaptic input (adapted condition, $+1.4 \pm 0.6$ pC; $p_{adapted} = 3.7 \times 10^{-2}$; unadapted condition, $+0.06 \pm 0.11$ pC; $p_{unadapted} = 0.47$; $n = 10$ cells; Wilcoxon signed rank test; Fig. 7e). This response pattern was consistent with a decrease in presynaptic inhibition following the offset of the adapting stimulus, resulting in an increase in glutamate release from midget bipolar cells. Thus, our recordings in the midget pathway of primate retina were consistent with the circuit motif proposed in other vertebrate species (Figs. 6 and 7f)[11].

**Sensitizing circuits more accurately reconstruct natural stimuli than adapting circuits.** We next sought to understand how these differing strategies of adaptation and sensitization impacted encoding during naturalistic vision. This was done by testing the ability of adapting and sensitizing models to accurately encode natural scenes. We specifically wanted to determine how accurately downstream visual circuits could reconstruct naturalistic input stimuli based on the outputs of populations of model On and Off midget ganglion cells. The naturalistic stimuli used in the

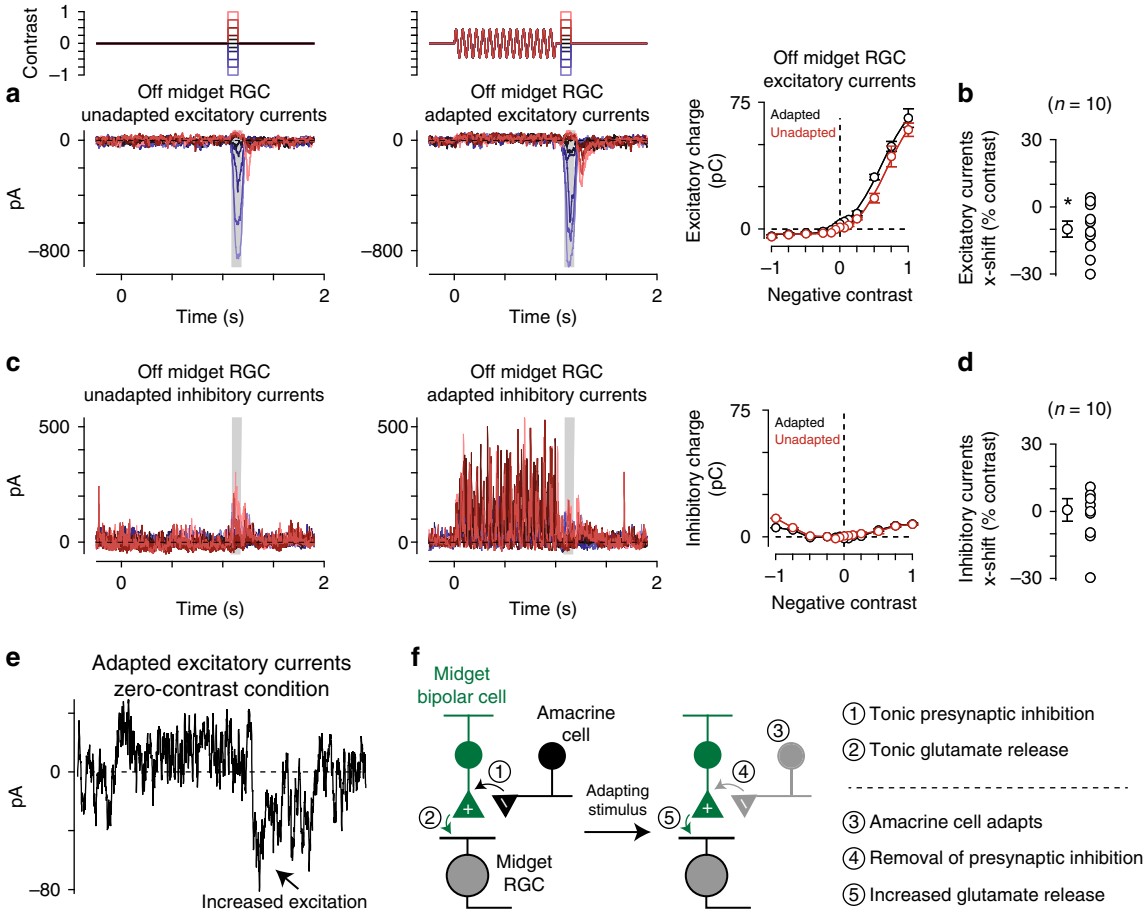

**Fig. 7** Sensitization was present in excitatory synaptic input from midget bipolar cells. **a** Excitatory synaptic currents from a central Off midget ganglion cell to a series of spots (diameter, 40–80 μm) centered over the receptive field. Spots were either presented alone (left) or 50 ms following the offset of an adapting stimulus (right; diameter, 730 μm). Shaded regions indicate sampling windows. Right: Average excitatory synaptic charge across the shaded regions. The wide-field adaptation evoked a leftward shift in the contrast-response curve (black) relative to the unadapted control condition (red). **b** Population data from midget cells showing the x-axis shift for adapted relative to unadapted conditions in excitatory synaptic currents. The adapting stimulus evoked a significant leftward shift relative to the unadapted condition, indicating that contrast sensitization was present in the excitatory input from midget bipolar cells to midget ganglion cells ($n = 10$ cells; $p = 1.4 \times 10^{-2}$). Mean values are shown in gray. Open circles are individual cells. **c** Inhibitory synaptic currents from the Off midget cell in (**a**). Inhibitory currents did not show a noticeable shift along the x-axis (right) and were small relative to excitatory currents recorded in the same cell (compare (**a**) with (**c**)). **d** Horizontal shift values for inhibitory synaptic inputs as in (**b**). Inhibitory inputs did not show consistent leftward shifts, indicating that postsynaptic inhibition was unlikely to contribute to the contrast sensitization observed in the spike output of midget ganglion cells ($n = 10$ cells; $p = 0.15$). Error bars indicate mean ± SEM. Statistical significance determined with Wilcoxon signed rank test. **e** Excitatory current recordings from the Off midget cell in (**a**) under the condition in which the stimulus intensity returned to the mean luminance after the offset of the adapting stimulus and an additional test flash was not presented (zero-contrast condition). A sustained increase in excitatory current was observed at the offset of that stimulus. **f** Proposed model for contrast sensitization in midget bipolar cells. Statistical significance was determined using the Wilcoxon signed rank test

model were taken from the DOVES database—a dataset of eye movements in humans recorded while observing natural images[28]. Reconstruction accuracy was determined by calculating the correlation between the stimulus and response of each model (see Methods). Periods of fixation between ballistic eye movements are critically important to visual coding in primates; thus, model performance was separately calculated for the complete movie or for periods of fixation only.

We considered two different decoding models for estimating the stimulus contrast based on the outputs of On and Off midget ganglion cells (see Methods). Regardless of the decoding scheme used, the sensitizing model showed higher accuracy for reconstructing the entire stimulus trajectory than either the adapting model or the LN model (Fig. 9c). The sensitizing model also outperformed the other models when the analysis was restricted to periods of fixation (Fig. 9d). We interpret the

outperformance of the sensitization model over the adaptation and linear–nonlinear models as support for the hypothesis that neural sensitization improves faithful encoding of visual inputs to midget ganglion cells.

We next sought insight into the stimulus conditions in which neural adaptation would improve encoding. Previous work supported a role for adaptation in determining when salient stimulus features changed[14,20,29]. Thus, we fit our models to the change in contrast as a function of time (see Methods). Indeed, the adaptation model outperformed both the sensitizing and linear–nonlinear models at reproducing the change in contrast (Fig. 9e). In fact, the adaptation model performed significantly better at encoding the change in contrast than at encoding the contrast itself—model correlation improved by 149 ± 10% for the linear decoding paradigm and 27 ± 2% for the quadratic decoding paradigm ($p < 3.9 \times 10^{-21}$; $n = 161$ movies; Wilcoxon signed rank

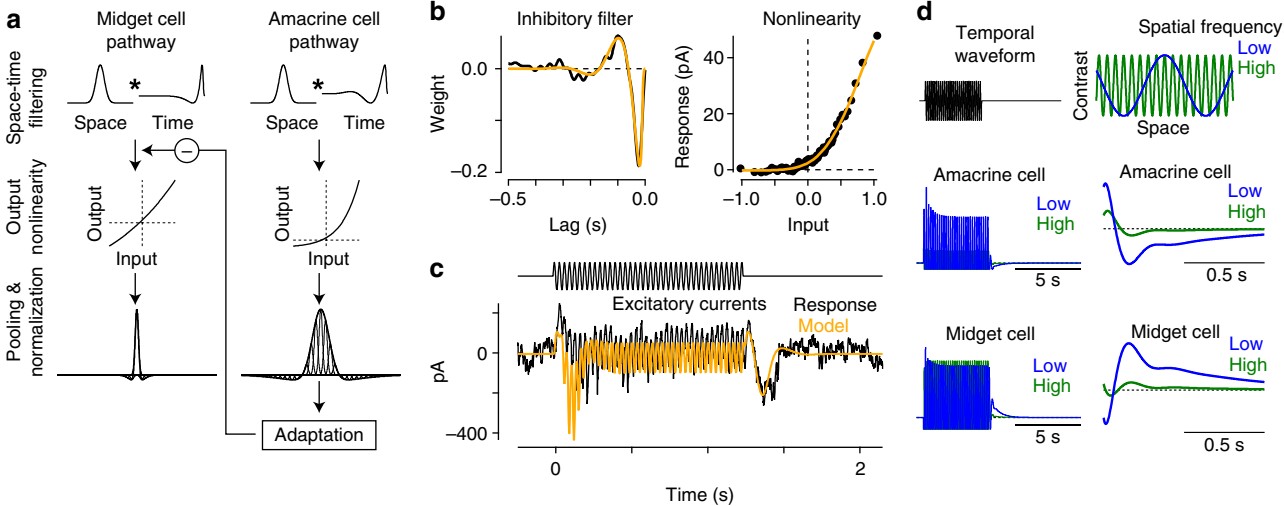

**Fig. 8** Sensitization model reproduces experimental results. **a** Sensitization model structure. Visual inputs were convolved with a spatiotemporal linear filter comprised of a Gaussian in space and a biphasic filter in time. Signals in the amacrine cell pathway were then passed through an output nonlinearity before passing to the adaptation stage of the model. The output of the amacrine cell model provided inhibitory input to the midget bipolar cell pathway upstream of the bipolar cell output nonlinearity. **b** Inhibitory temporal filter (left) and input–output nonlinearity (right) determined from noise recordings. These filters were then used as components of the computational model (**a**). **c** Excitatory current recording from an Off midget ganglion cell to the wide-field adapting stimulus (see Fig. 7). Model prediction (orange) was generated from excitatory synaptic current recordings to the noise stimulus in the same cell. **d** Model output for drifting grating stimuli at high and low spatial frequencies

test). Possible benefits of representing the temporal derivative of contrast in natural vision are considered in the Discussion.

**Background motion evokes contrast sensitization in midget cells**. The finding that the sensitization model outperformed the other model paradigms during periods of fixation suggested that sensitization could play a particularly important role in vision during periods of fixation following the offset of global motion. Mechanisms that would maintain sensitivity during eye movements would be particularly critical given the high frequency of these movements in primates. We, thus, sought to determine whether background motion could evoke contrast sensitization with direct recordings from midget ganglion cells. We measured contrast responses in parasol and midget cells following the offset of a full-field moving texture (speed, 5–11 degrees s⁻¹; duration, 1 s). The goal was to simulate, as closely as possible, the brief periods of fixation following eye movements and to test sensitivity during these fixation periods. We interleaved these recordings with measurements when the texture was stationary throughout the trial.

As with the other adapting stimuli, background motion elicited a reduction in sensitivity in parasol cells (Fig. 10). In midget cells, however, background motion elicited increased sensitivity at low contrast (Fig. 10f). These results indicated that eye motion should produce profoundly different effects on the two numerically dominant output pathways of the primate retina. As with the computational model (Fig. 9), adaptation in parasol cells would render them less sensitive following eye motion. Due to sensitization, however, the midget pathway would be poised to report information about a fixated object following eye movements.

## Discussion

Our results support a novel role for neural sensitization in primates relative to the function proposed in other species. Sensitizing cells are commonly thought to counteract the loss of responsiveness experienced by adapting cells during transitions from high to low variance environments[10]. This hypothesis requires that sensitizing cells have an adapting counterpart that

encodes similar information about the environment. Midget (parvocellular-projecting) ganglion cells are well known for their roles in both chromatic and achromatic vision[21–23,30]. Functional parallelism in the midget pathway is achieved by splitting signals between different classes of cone photoreceptor (L versus M) or bipolar cell (On versus Off) inputs to the midget cell receptive field. Further, we found that both On- and Off-type midget cells exhibited sensitization (Figs. 1–3 and 10), and the primate retina lacks an adapting functional counterpart to midget cells with similar chromatic opponency or spatial acuity[31]; thus, sensitization does not counterbalance adaptation in another functionally parallel pathway. This conclusion is further bolstered by the observation that the spike output of midget cells did not saturate during high variance stimulation (Fig. 5), indicating the sensitization performs a distinct function in primate retina relative to other species[10].

Instead, our findings support a role for sensitization in maintaining the responsiveness of the midget pathway during dynamic visual processes, such as head or eye movements, that cause rapid fluctuations in light intensity on the retina. We base this conclusion on several key observations. First, sensitization was strongest following wide-field stimulation (Figs. 1 and 2) or background motion (Fig. 10). Second, sensitization persisted for >0.2 s (Fig. 3), a period that roughly corresponds to the durations of fixations following eye movements in primates (reviewed in ref. 32). Finally, sensitization greatly improved the fidelity of encoding natural movies, particularly during periods of fixation following ballistic eye motion (Fig. 9). Thus, sensitization appears to play a unique and crucial role in neural coding in primates.

A parallel study also found evidence supporting the link between the sensitization mechanisms that we observed in midget ganglion cells and visual perception in humans[33]. Subjects showed a significant enhancement in contrast sensitivity following the offset of wide-field motion; and this increase in sensitivity was manifest as a leftward horizontal shift in the perceptual input–output relationship, just as we observed in midget cells (compare Fig. 2 in our study with Fig. 5 of ref. 33). Together, these findings provide a rare example of a behavior that can be directly tied to a specific neural circuit motif.

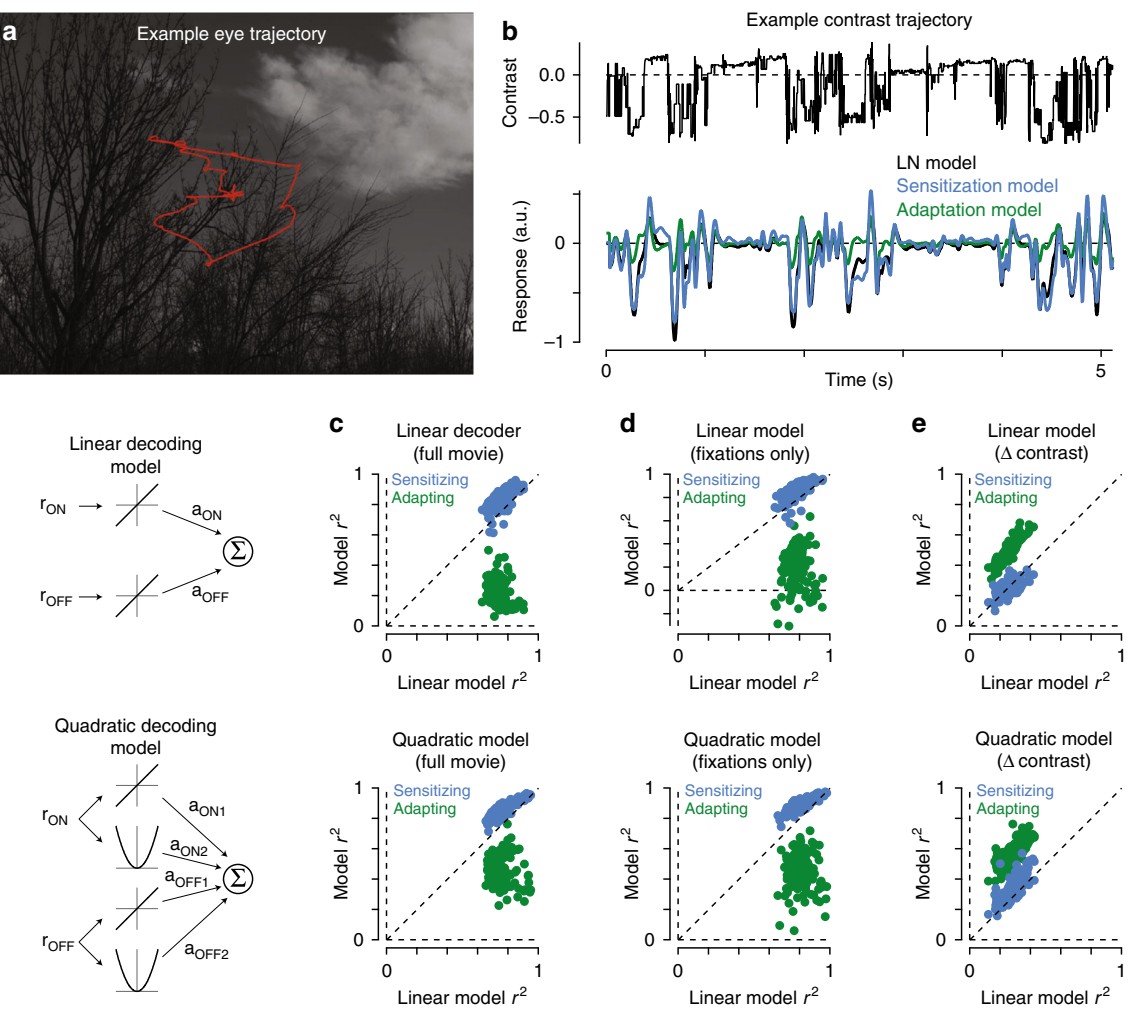

**Fig. 9** Sensitization increases the fidelity of encoding natural movies. **a** Example image from the DOVES database. The observer's eye trajectory is shown in red. **b** Top: temporal contrast sequence from the eye movement data in (**a**). Bottom: responses of the adaptation and sensitization models to the example contrast sequence. **c** Performance of the sensitization and adaptation (y-axis) models at reconstructing 161 natural movies in the database. Model performance is shown relative to the linear–nonlinear model performance for each movie (x-axis). Performance was measured as the Pearson correlation between the stimulus and model predictions after adjusting for temporal lag. Performance for each movie is indicated by a dot. The sensitization model outperformed the linear–nonlinear and adaptation models ($p < 6.2 \times 10^{-26}$). **d** Model performances as in (**c**), but restricted to periods of fixation. The sensitization model again outperformed the linear–nonlinear and adaptation models ($p < 4.4 \times 10^{-25}$). **e** Model performance for the change in contrast as a function of time. The adaptation model outperformed both the linear–nonlinear and sensitizing models at decoding the change in the contrast trajectory ($p < 1.0 \times 10^{-23}$). Statistical tests were paired and were determined using Wilcoxon signed rank test

**Distinct functions of adaptation and sensitization in primate retina.** Our findings also speak to the roles of neural adaptation in the parasol and broad thorny ganglion cell pathways. Previous work proposed that adapting cells could produce a nearly optimal faithful encoding of sensory inputs[20]. Our computational model, however, indicates that sensitizing circuits outperform adapting circuits in faithfully encoding natural movies (Fig. 9). The improved reconstruction accuracy of the sensitizing model was consistent with a recent theoretical report indicating that sensitizing cells are better for encoding faithful representations of sensory input than adapting cells[14]. According to this paradigm, sensitizing cells such as midget ganglion cells would be useful for directly encoding information about the properties of the input (e.g., contrast, color). Adapting cells, on the other hand, are optimized for performing inference tasks[14,29].

Adapting cells dynamically adjust their input–output properties to align with the recent stimulus distribution[4,8]. These adjustments make the cells exquisitely sensitive to changes in stimulus statistics, allowing them to infer when salient properties of the environment change. For example, quickly detecting object motion is an ethologically relevant and phylogenetically ancient neural computation[34,35]; by decreasing their responsiveness during periods in which the background is either stationary or coherently moving, adapting neural circuits would be poised to report when an object moves relative to the background[36,37]. Indeed, our adapting model outperformed other models at representing the change in contrast as a function of time (Fig. 9). Algorithms that compute changes in image contrast as a function of time are commonly used in computer-based vision systems that calculate visual motion features such as optical flow. Whether the output of adapting retinal cells, such as parasol cells or broad thorny cells, might also be used for similar computations by downstream visual circuits is not currently known. However, both parasol and broad thorny ganglion cells have been implicated in motion processing[25,37–39] and they project to retinorecipient brain regions in the lateral geniculate body, superior colliculus, and inferior pulvinar that contribute significantly to motion vision[40–42].

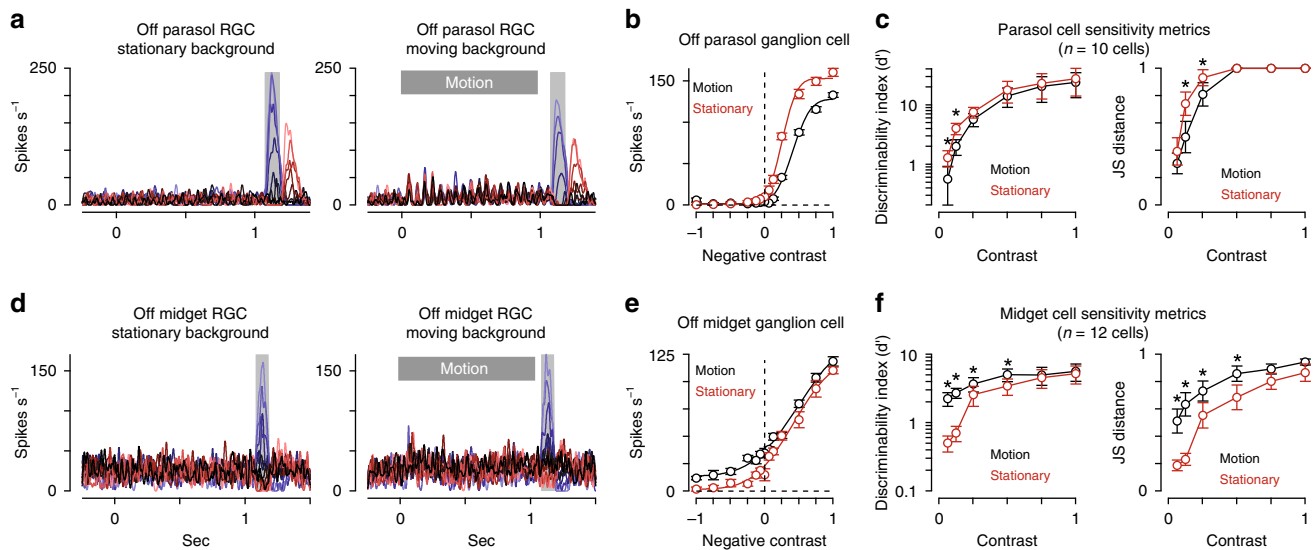

**Fig. 10** Background motion evokes adaptation in parasol cells and sensitization in midget cells. **a** Averages spike rate as a function of time for an Off parasol ganglion cell a stationary texture followed by a series of test flashes centered over the cell's receptive field (left) or following the offset of texture motion (right; speed, 11 degrees s$^{-1}$). **b** Contrast-response functions for the cell in (**a**) for the measurements with a stationary texture (red) or a moving texture (black). **c** Sensitivity metrics for this experiment across 10 parasol cells. **d** Spike responses from an Off midget ganglion cell to the same experimental protocol. **e** Average spike rate across the shaded regions indicated in (**d**). The wide-field adaptation evoked a leftward shift in the contrast-response curve (black) relative to the unadapted control condition (red). **f** Sensitivity values for the motion experiment across 12 midget cells. Motion produced a significant increase in sensitivity at low contrast relative to the stationary condition (contrast, ≤50%; $p < 0.05$; Wilcoxon signed rank test). Circles and bars indicate mean ± SEM

**Relationship to psychophysical measurements in humans**. It has long been recognized that eye movements play important computational roles in visual processing (reviewed in refs. [43,44]). Periods in which an image is stabilized on the retina cause that image to fade from perception[45] and small fixational eye movements appear to counteract this fading[46,47]. These eye movements can, however, produce large temporal fluctuations in contrast, particularly when viewing high-contrast objects. This would, in turn, produce fading phenomena in cells that strongly adapt, such as parasol ganglion cells—a prediction that was confirmed with our background motion experiments and computational model (Figs. 9 and 10).

Neural mechanisms such as sensitization may serve to counteract adaptation by maintaining the sensitivity of certain visual pathways during eye movements. Indeed, our computational model and direct measurements indicated that contrast sensitization in the midget ganglion cell pathway was engaged well by background motion such as that observed during eye movements (Figs. 9 and 10). Thus, contrast sensitization might act to maintain sensitivity of image-forming visual pathways following eye movements that are commonplace in primate vision. Indeed, psychophysical studies in humans indicated that contrast sensitivity increases following both ballistic (saccade) and fixational eye movements[46–49]. This increase in sensitivity was observed to chromatic stimuli and high-spatial-frequency achromatic stimuli, mirroring our results in midget ganglion cells. It was hypothesized that these increases in sensitivity arose in the thalamus[47]; however, our results support an origin much earlier in the visual pathway—in the interplay between wide-field amacrine cells and midget bipolar cells in the inner retina (Fig. 7).

**Future directions**. Given its recent discovery, relatively few studies have investigated the role of sensitization in neural processing[10–12,14,16]. Here, we used a complementary set of visual stimuli and a computational model in our attempts to understand this phenomenon, but many questions remain. For example, the roles that sensitization plays during natural vision are not well understood, and we developed a computational model to gain insight into this question. The parameters for this model were determined from direct recordings of synaptic inputs to midget ganglion cells using uncorrelated noise. Recent work, however, has clearly demonstrated that the recruitment of different neural mechanisms can vary dramatically between artificial and naturalistic stimuli[50–52]. Thus, studies of adaptation and sensitization in the context of more naturalistic stimuli are needed to elucidate the roles of short-term plasticity mechanisms in visual processing.

## Methods

**Tissue preparation and electrophysiology**. Experiments were performed in an in vitro, pigment-epithelium attached preparation of the macaque monkey retina[53]. Eyes were dissected from terminally anesthetized macaque monkeys of either sex (Macaca *fascicularis*, *mulatta*, and *nemestrina*) obtained through the Tissue Distribution Program of the National Primate Research Center at the University of Washington. All procedures were approved by the University of Washington Institutional Animal Care and Use Committee.

The retina was continuously superfused with warmed (32–35 °C) Ames' medium (Sigma) at ~6–8 mL min$^{-1}$. Recordings were performed from macular, mid-peripheral, or peripheral retina (2–8 mm, 10–30 foveal eccentricity), but special emphasis was placed on recording from more centrally located cells. Physiological data were acquired at 10 kHz using a Multiclamp 700B amplifier (Molecular Devices), Bessel filtered at 3 kHz (900 CT, Frequency Devices), digitized using an ITC-18 analog-digital board (HEKA Instruments), and acquired using the Symphony acquisition software developed in Fred Rieke's laboratory (http://symphony-das.github.io).

Recordings were performed using borosilicate glass pipettes containing Ames medium for extracellular spike recording or, for whole-cell recording, a cesium-based internal solution containing (in mM): 105 CsCH$_3$SO$_3$, 10 TEA-Cl, 20 HEPES, 10 EGTA, 2 QX-314, 5 Mg-ATP, and 0.5 Tris-GTP, pH ~7.3 with CsOH, ~280 mOsm. Series resistance (~3–9 MΩ) was compensated online by 50%. The membrane potential was corrected offline for the approximately –11 mV liquid junction potential between the intracellular solution and the extracellular medium. Excitatory and inhibitory synaptic currents were isolated by holding midget ganglion cells at the reversal potentials for inhibition/chloride (~–70 mV) and excitation (0 mV), respectively.

**Visual stimuli and data analysis**. Visual stimuli were generated using the Stage software package developed in the Rieke lab (http://stage-vss.github.io) and

displayed on a digital light projector (Lightcrafter 4500; Texas Instruments) modified with custom LEDs with peak wavelengths of 405, 505 (or 475), and 640 nm. Stimuli were focused on the photoreceptor outer segments through a ×10 microscope objective. Mean light levels were in the low to medium photopic regimes ($\sim 3 \times 10^3$–$3.4 \times 10^4$ photoisomerizations [R*] cone$^{-1}$ s$^{-1}$). Contrast values for contrast-response flashes are given in Weber contrast and for periodic stimuli in Michaelson contrast. All responses were analyzed in MATLAB (R2018b, Mathworks).

For extracellular recordings, currents were wavelet filtered to remove slow drift and amplify spikes relative to the noise[54] and spikes were detected using either a custom k-means clustering algorithm or by choosing a manual threshold. Whole-cell recordings were leak subtracted and responses were measured relative to the median membrane currents immediately preceding stimulus onset (0.25–0.5 s window).

**Sensitivity calculations**. We evaluated a cell's ability to accurately detect a change in contrast using two independent measures—the discriminability index and the Jensen-Shannon distance. The discriminability index ($d'$) is a relatively simple metric for determining the amount of overlap between signal and noise distributions:

$$d' = \frac{\mu_S - \mu_N}{\sqrt{\frac{1}{2}(\sigma_S^2 + \sigma_N^2)}} \quad (2)$$

where $\mu_S$ and $\mu_N$ are the mean of the signal and noise distributions and $\sigma_S^2$ and $\sigma_N^2$ are the variances of those distributions, respectively.

The Jensen-Shannon distance was calculated from the Kullback-Leibler divergence ($D_{KL}$) between the spike count probability distributions for the signal ($P$) and noise ($Q$).

$$JS(P, Q) = \frac{1}{2}\left[D_{KL}\left(P\left|\frac{P+Q}{2}\right.\right) + D_{KL}\left(Q\left|\frac{P+Q}{2}\right.\right)\right] \quad (3)$$

The Kullback-Leibler divergence between the spike count distributions was calculated as:

$$D_{KL}(P||Q) = \sum_n p_n \log_2\left(\frac{p_n}{q_n}\right) \quad (4)$$

where $p_n$ is the probability of observing $n$ spikes in the sample window and $q_n$ is the probability of observing $n$ spikes in the sample window during presentation of a uniform mean background.

**Temporal noise analysis**. To directly measure how changes in stimulus variance affected temporal filtering and sensitivity, we presented a Gaussian flicker stimulus. Equivalent periods of high and low variance were presented on each trial, and separate temporal filters were calculated for these periods by cross-correlating the contrast trajectory ($S$) with the cell's spike output ($R$)[4].

$$F(t) = \int R(\tau)S(t + \tau)d\tau \quad (5)$$

The resulting filters were fit with a function commonly used to model time-domain filtering of retinal cells[55,56]:

$$F(t) = A\frac{(t/\tau_{\text{rise}})^n}{1 + (t/\tau_{\text{rise}})^n}e^{-(t/\tau_{\text{decay}})}\cos\left(\frac{2\pi t}{\tau_{\text{period}}} + \varphi\right) \quad (6)$$

where $A$ is a scaling factor, $\tau_{rise}$ is the rising-phase time constant, $\tau_{decay}$ is the damping time constant, $\tau_{period}$ is the oscillator period, and $\varphi$ is the phase (in degrees).

The input–output nonlinearity was calculated by convolving the temporal filter and stimulus to generate the linear prediction. The prediction ($x$-axis) and response ($y$-axis) were modeled as a cumulative Gaussian distribution[57].

$$N(x) = \varepsilon + \frac{\alpha}{\sqrt{2\pi}}\int_{-\infty}^{x} e^{\frac{-(\beta t + \gamma)^2}{2}}dt \quad (7)$$

where $\alpha$ indicates the maximal output value, $\varepsilon$ is the vertical offset, $\beta$ is the sensitivity of the output to the generator signal (input), and $\gamma$ is the maintained input to the cell. Input–output nonlinearities were separately calculated for three distinct stimulus–response regions: (1) the period of high contrast stimulation, (2) the period of low-contrast stimulation immediately following the transition from high contrast (100–600 ms; low early), and (3) the sustained period of low contrast (>1 s following the high-to-low transition; low late).

Changes in sensitivity can result in changes in the maximal slope (i.e., gain) or horizontal shifts in this input–output nonlinearity. Thus, we simultaneously fit the high and low contrast filters such that the gain and horizontal offset were allowed to vary between the filters and the other parameters were shared[18,58]. Fitting was performed via nonlinear least-squares curve fitting.

To evaluate model performance, we interleaved trials in which a unique contrast trajectory was presented to a cell with trials in which the contrast trajectory was not unique (noise seed = 1). These non-unique trials were equally interspersed with the unique trials. Model performance was evaluated by averaging the responses from non-unique trials and calculating the Pearson correlation coefficient between the model prediction and this average response.

**Sensitization and adaptation models**. We modeled spatiotemporal integration in bipolar cells and amacrine cells as the product of a Gaussian spatial filter and a biphasic temporal filter which was then passed through an input–output nonlinearity. The output of this nonlinear stage of the amacrine cell model was then passed through an adaptation stage; adaptation in the amacrine cell provided inhibitory input to the bipolar cell model prior to the output nonlinearity (Fig. 8a). Following the subunit output, model midget ganglion cells and amacrine cells pooled (summed) inputs from bipolar cell subunits and the weights of these inputs were normalized by the subunit location relative to the receptive-field center using a Gaussian weighting.

To estimate the excitatory and inhibitory circuit components for the computational model, we recorded excitatory and inhibitory synaptic currents from midget ganglion cells in response to a full-field Gaussian flicker stimulus. The contrast of each frame was drawn randomly from a Gaussian distribution and that value was multiplied by the average contrast. Average contrast was updated every 0.5 s and drawn from a uniform distribution (0.05–0.35 RMS contrast). The linear temporal filters ($F$) were calculated by cross-correlating the stimulus sequence ($S$) and the leak-subtracted response ($R$) as described above.

The spatial component of the bipolar and amacrine cell receptive fields was modeled as a Gaussian function with a 2-SD width of 18 and 90 μm, respectively. Each midget ganglion cell was modeled as receiving input from a single bipolar cell, as is typically the case in the central retina. Sensitization parameters were determined by fitting linear–nonlinear model predictions relative to the excitatory currents recorded to the Gaussian flicker stimulus.

The amacrine cell providing direct inhibition to the midget ganglion cells is likely distinct from the cell providing presynaptic inhibition at the level of the midget bipolar cell (see Fig. 7). Thus, our inhibitory synaptic recordings likely did not grant us direct access to the properties of the amacrine cell responsible for contrast sensitization. These recordings do, however, provide an estimate of the time course of signals passing through the presynaptic amacrine cell to midget bipolar cells. Signals passing through this amacrine cell proceed from cone photoreceptors to bipolar cells and then to the amacrine cell in question before providing input to the midget bipolar cell. In the same way, the amacrine cell providing direct inhibition to midget ganglion cells must pass through an extra synapse. Thus, our recordings of direct synaptic inhibition were useful in approximating the time course (i.e., temporal lag) of presynaptic inhibition at the midget bipolar terminal.

**Evaluating model performance to naturalistic movies**. We evaluated the performance of the adaptation and sensitization models in reconstructing the naturalistic movie sequences using linear and quadratic decoding paradigms. To estimate stimulus contrast, the linear decoder ($f_{\text{LINEAR}}$) summed the scaled outputs of the model On and Off midget ganglion cells:

$$f_{\text{LINEAR}}(t) = a_{\text{ON}}r_{\text{ON}}(t) + a_{\text{OFF}}r_{\text{OFF}}(t) + k \quad (8)$$

where $a_{\text{ON}}$ and $a_{\text{OFF}}$ are scaling constants and $k$ is an offset constant. The quadratic model was similar in structure except that the response from each pathways was squared prior to summation:

$$f_{\text{QUADRATIC}}(t) = a_{\text{ON1}}r_{\text{ON}}(t) + a_{\text{ON2}}r_{\text{ON}}^2(t) + a_{\text{OFF1}}r_{\text{OFF}}(t) + a_{\text{OFF2}}r_{\text{OFF}}^2(t) + k \quad (9)$$

For each of the 161 movies in the database, the input stimulus was shifted to the peak of the midget temporal filter (~35 ms) and then scaling and offset coefficients were determined using least-squares curve fitting. The Pearson correlation was then calculated between the temporal trajectories of the model and the movie.

We used the same technique to also evaluate model performance in reconstructing the change in contrast as a function of time. The change in contrast was calculated by taking the first derivative of the contrast trajectory and low-pass filtering the resulting vector with a Gaussian filter (s.d., 20 ms).

**Quantification and statistical analysis**. All statistical analyses were performed in MATLAB (R2018b, Mathworks). Reported $p$ values in this study were either determined using the Wilcoxon signed rank test for paired data and the Wilcoxon rank sum test (i.e., Mann–Whitney $U$ test) for unpaired data. Final figures were created in MATLAB, Igor Pro, and Adobe Illustrator.

**Reporting summary**. Further information on research design is available in the Nature Research Reporting Summary linked to this article.

## Data availability
The datasets generated during and/or analyzed during the current study are available from the corresponding author on reasonable request.

## Code availability

Visual stimulation (http://stage-vss.github.io) and data acquisition (http://symphony-das.github.io) software are freely available.

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

## Acknowledgements

The authors thank Shellee Cunnington, Mark Cafaro, and Jim Kuchenbecker for technical assistance. Tissue was provided by the Tissue Distribution Program at the Washington National Primate Research Center (WaNPRC; supported through National Institutes of Health grant P51 OD-010425), and we thank the WaNPRC staff, particularly Chris English, for making these experiments possible. Fred Rieke, Raunak Sinha, Max Turner, and Will Grimes assisted in tissue preparation. We thank Alison Weber and Fred Rieke for helpful discussions. We also thank Alison Weber and Jon Demb for feedback on a previous version of this manuscript. This work was supported in part by grants from

the National Institutes of Health (NEI R01-EY027323 to M.B.M.; NEI P30-EY001730 to the Vision Core), Research to Prevent Blindness Unrestricted Grant (to the University of Washington Department of Ophthalmology), Latham Vision Research Innovation Award (to M.B.M.), and the Alcon Young Investigator Award (to M.B.M.).

## Author contributions

Conceptualization, M.B.M.; Methodology, M.B.M.; Software, M.B.M.; Formal analysis, M.B.M.; Investigation, M.B.M. and T.R.A; Resources, M.B.M.; Data curation, M.B.M. and T.R.A.; Writing—original draft, M.B.M.; Writing—review & editing, M.B.M. and T.R.A.; Visualization, M.B.M.; Supervision, M.B.M.; Project administration, M.B.M.; Funding acquisition, M.B.M.

## Additional information

**Competing interests:** The authors declare no competing interests.

