## [Peer Review File · Nature Communications]

Reviewers' Comments:

Reviewer #1:

Remarks to the Author:

This study investigates the sensitization patterns of well-defined retinal ganglion cell (RGC) types of the macaque monkey retina. The authors show that midget, but not parasol or broad thorny RGCs, show increased firing after an adaptation period with high contrast grating stimuli. They then investigated the underlying mechanisms using whole-cell patch clamp recording and show that the sensitization of the midget cell spiking response is accompanied by the sensitization of their excitatory inputs. Furthermore, the authors used computational modeling to evaluate the role of this phenomenon in the encoding of natural scenes and found that the sensitizing model shows higher encoding accuracy compared to the adapting model, suggesting an advantage of sensitization in the midget pathway for visual processing.

Overall, this study addresses an important and interesting question, and the experiments were beautifully done in a technically challenging prep. My main concerns are listed below:

Specific comments:

1. It's not obvious from Figure 2 B and 2C that the increased firing rate immediately after adaptation reflects an increase in the contrast sensitivity of the midget cell. The adapted curves in Fig 2B and 8B appear to have an upward shift along the Y axis, suggesting an elevated baseline firing. This is evident by the enhanced spiking at zero contrast. In addition, the gain of the midget cell response is reduced after adaptation as measured by the slope of the curves in Figure 2B, consistent with a reduction in contrast sensitivity. Therefore, the higher firing rate of midget cells after adaptation might result from elevated firing at all contrasts (an upward shift) instead of enhanced contrast sensitivity (a leftward shift). How is "contrast sensitivity" defined in this study? Have the authors considered ways to separate the contrast response from the baseline firing at zero contrast?
2. The claim about the wide-field amacrine cell in the abstract seems too strong, because the evidence on the cell type(s) that confers sensitization of bipolar cell inputs is rather indirect and needs substantiation. I suggest a more speculative mention of this hypothesis.
3. According to Figure 5c, there seems to be both a leftward shift of excitation and a rightward shift of inhibition. Could inhibitory mechanism also contribute to the sensitization? It will be helpful to include a discussion about this possibility.
4. In the text about the different degree of shifts during wide-field and small-field test flashes, the authors mentioned that "this trend held true across midget cells-horizontal shifts were more negative for the small-diameter test flash than for the wide-field test flash in the same cell... (page 7)". However, in Fig. 2E, there seems to be an On/Off difference: this trend seems robust for On midget cells, but most of the Off midget cells (4 out of 5) do not show this trend. Please clarify. In addition, to support the above claim, a statistical analysis to directly compare the shifts under small and wide field conditions are necessary. It would be helpful to increase the n for this figure, though it is a big ask given the challenges of recording from monkey retinas.
5. The sensitizing model showed higher encoding accuracy for periods of fixation relative to periods of ballistic eye movements. However, this cannot be an argument to infer that "sensitization could play a particularly important role in vision during periods of fixation following the offset of global motion". To support this argument, the authors need to show that the improved performance of the sensitization model over adaptation or no-plasticity model (Δr^2) is larger during fixation than that during ballistic eye movements.
6. The sensitization model reproduces experimental results during grating stimuli in Figure 6. However, RGC receptive field properties may differ significantly during artificial (such as drifting gratings) and natural stimuli, and they may rely on different modes of spatial integration (for example, Turner et al 2018; Turner and Rieke 2016). Therefore, this model may not accurately predict

midget cell responses during natural stimuli. It will be helpful to discuss this potential issue in the discussion.

Minor comments:

1. Please indicate statistical significance in the figures and list the sample sizes and p values in the figure legends (e.g Fig.1e, Fig.2c and 2e, Fig. 4d, Fig. 5c, Fig. 8c and 8f) so it would be easier for the readers to find it and comprehend the conclusions.
2. Page 4, line 87: "At the transition from high to low contrast", should it be changed to "... to no contrast" since the schematics in Fig 1A indicate there's no contrast?
3. Please clarify in the Methods how the x-shift values are calculated from two curves (e.g Fig.2A, 2B and 2D).
4. Line 71: "to determine whether short-term plasticity in the midget pathway depended on ...": should be "depends on"
5. Text line 161 indicates the contrast as ± 0.5 while figure 3 legend indicates the contrast as $\pm 0.5 - 0.5$.
6. Figure 6B is not referenced in the main text.

Reviewer #2:

Remarks to the Author:

Sensory neurons adapt to the variance of the input by adjusting their gain and threshold. Classically, ganglion cells were reported to adapt when there is a switch from high variance to low variance by progressively increasing their gain and lowering their threshold. In the transient part right after this switch, the neuron is still adapted to the high variance, and this leads to a very low firing rate and probably a loss in information.

More recently, several works have shown that there is another type of behaviour among ganglion cells, i.e. sensitizing: firing rate increases following the switch and gain is higher. This could compensate for the loss of information, but it could also be that sensitizing cells are not just here to complement classically adapting cells.

The present paper aims at contributing to this in a meaningful way: first, by reporting for the first time sensitizing behaviour in the primate retina. Second, by showing that sensitizing is present in midget ganglion cells, while parasol cells are adapting. This is interesting because these two types compute very different features and are not functional counterparts. This goes against the idea of complementing adapting cells.

Since this is the first report of sensitizing ganglion cells in primate, I think this paper is of significance, especially with recent studies now speculating about the perceptual consequences of sensitization.

However, I have several major concerns that need to be addressed to make this study really convincing:

1) first, it is not clear that the behaviour they found is similar to what was found in the salamander retina. As a consequence, it is not clear if we are really talking about the same thing or not. In particular, the authors report here a small decrease in gain and a change in threshold. Kastner and Baccus, in their seminal paper about contrast sensitization, report an increase in gain. This is an important difference and it is not clear if the analogy still holds. See also my comments below about other differences.

Also, the stimuli that are used here are different from the ones in this previous study and this difference should at least be discussed.

2) second, the title and some claims of the abstract are a bit misleading, since they suggest that the sensitization "improves encoding". This means that the mutual information between the stimulus and the response is higher than if there was no sensitization, but this is never shown in the data. The only place where this issue is tackled is in the model, but it is hard to make a convincing case using only a model whose agreement with data is only qualitative (it is never used to predict ganglion cell responses directly), and noise is not modeled carefully. If the authors want to make this claim, estimating mutual information on the data is necessary (and possible).

3) third, the number of cells recorded is low, especially since there is a significant variability in the reported population data (fig2 and even more fig 4 and 5). Also, it seems that ON and OFF cells should be separated in the analysis. It could be that just one of them is sensitizing, there is no good justification to pool them together.

I have a list of additional concerns which should also be addressed.

-fig1-2, experiment with grating.

Similar to Kastner and Baccus averaging across multiple noise instantiations, it would be better to average firing rate over different stimuli where the initial and final phase of the grating is picked at random. This would also exclude that the transient increase in firing rate is not just a slow ON or OFF response to the last grating presented, or anything phase-dependent.

-fig 2: color labeling is strange, why not a different color for each contrast level ?

It is not clear how gain is estimated here - methods should be more explicit about this: maybe they fit a sigmoid curve and report the change in parameters? The way it is reported, x-shift, gain etc, is a bit confusing and might be difficult to follow for the naive reader.

-fig 3: there is some possible confusion in the use of the terms wide field and narrow field, and this confusion is present in the discussion too. My understanding is that the best way to have sensitization is a wide field noise followed by a local probe stimulus, is it correct ? Please clarify this.

-One can also regret that there is no study of how the sensitization depends on the stimulus location, similar to Kastner and Baccus 2013. Maybe it remains adaptive when the probe stimulus is in the surround, and this explains why the sensitization is weaker in the surround ? Or is it weaker just because of surround suppression ? I don't think a full study on this can be expected, but at least some discussion would be welcomed.

-fig 3: what is the window over which firing rate is estimated ? Is delay defined by the starting point of this window ? When is it significantly different from 0 ? More description of the methods used would be welcomed.

Also, error bars are large and n quite small (see above).

Axis legend: I guess log delay means delay with a log scale ? A bit of a misnomer, just "delay" would be better.

-fig5D: there should be population analysis at least in the corresponding text.

-Fig 5: please show the inhibitory counterpart fully with an example of raw data, not just 5C.

-fig 6: the model predicts adaptation of the amacrine output: do you see it in inhibitory current

recordings ?

-fig7: I don't think the modeling study is very insightful as it stands. The purpose of this kind of modeling should rather be to understand what make adapting cells worse at encoding the stimulus, but this study provides no insight about this.

Note also that Kastner and Baccus reported a saturation of of the non-linearity in high variance stimuli for sensitizing cells in salamander. For this reason, adapting cells encoded better the stimulus during episode of high variance. Anything similar here ? Answering this question would be a great way to exclude (or not) the hypothesis of complementary pathways.

Maybe the model can also be used to explore the range of statistics where sensitizing cells encode better. Here we are left with the impression that they also do a better job at this than adapting cells, even if the discussion says that adapting cells are better "to infer when salient properties of the environment change". If you are always better at encoding the stimulus is not clear how you will be worse at detecting motion. Please clarify.

-fig 8: no increase in activity similar to figure 2. Why ? The model would predict such an increase. Actually, if anything, there might be a weak increase for parasol cells, but maybe not significant.

Eccentricity of the recorded cells should also be reported, as it could affect the results and their interpretation significantly, especially regarding to connection to psychophysics.

Reviewer #3:

Remarks to the Author:

This paper demonstrates and explains the mechanisms underlying sensitization of visual responses in midretinal ganglion cells of the primate retina. The authors show that a wide-field, low spatial frequency stimulus reduces subsequent responses in parasol cells (i.e., adaptation) but enhances responses in midretinal ganglion cells (facilitation). Facilitation (or "sensitization") lasts for 100's of ms and is greatest when the test stimulus is small and centered in the middle of the cell's receptive field. The authors go on to show very nicely that the circuit is sensitized following achromatic stimulation, arguing against a significant role for horizontal cell-cone feedback mechanisms. Sensitization is present in the excitatory synaptic inputs to midretinal ganglion cells, indicating that it is not due to postsynaptic integration of excitatory and inhibitory inputs. Instead, the results point to lateral feedback inhibition from amacrine cells onto presynaptic bipolar cell terminals as the primary mechanism. Final experiments show that sensitization occurs in response to background motion similar to a saccadic eye movement, adding to the idea in the field that inhibitory circuitry helps the retina account for self-motion. A mathematical model nicely suggests that sensitizing circuitry likely does a better job reconstructing natural stimuli than adapting circuitry.

This is a very nice paper. The experiments are carefully done, nicely analyzed and (for the most part) clearly presented. All of my comments, save one, are simple suggestions to slightly enhance the clarity of the presentation.

There is a lot of work in this paper, and I hesitate to ask for another experiment, so I hope the authors (and editors) will take this as a suggestion rather than a requirement. The authors show that lateral inhibition underlies sensitization. It would be interesting to examine the spatial scale of this inhibition, i.e., how does sensitization depend on the diameter of annular surround stimulation? How does this spatial scale compare to that of presynaptic inhibition as detected in EPSCs recorded in the

midget ganglion cells? As the field anticipates a primate retinal connectome, this information would be invaluable for subsequently identifying the amacrine cell(s) mediating the sensitizing mechanism.

Minor comments:

In the Results, you describe the midget data in figure 1 prior to the parasol/thorny data, yet the figure is organized in the opposite order. It seems more logical to place the midget data in panels A and B.

Page 9: The description of the results in Figure 4 is well-written, but it would be helpful to the reader if the authors would make more specific references in the text to relevant panels in the figure.

Line 314: "Differencing": I was surprised to find that this is actually a word, at least in the context of heraldry, but I think "subtracting" is the better choice.

We thank the reviewers for their excellent insights and evaluation of this paper. In response to the reviewers' feedback, we have performed additional experiments, analyses, and updates to the text. These additions are highlighted in the revised manuscript and inline with the reviewers' comments, below. We sought to address each of the reviewers' comments with new experiments and analysis. The reviewers expressed two principal concerns. The first concern, summarized by the editor, was regarding the "low number of recorded cells." We have addressed this concern by substantially increasing the number of recorded cells in all of the experiments for the manuscript. In addition, we have added two major experiments, based on reviewer feedback, with many more cells and these experiments further bolster the main findings of the original work. The recorded cells for this manuscript are nearly all from central regions of the macaque retina, which greatly increased the technical challenge of obtaining high quality recordings. Given the clear link between our findings and psychophysical and behavior studies of humans and non-human primates, we felt it necessary to focus our experiments in central regions of the retina.

Secondly, all three reviewers wanted us to clarify the spatial extent of contrast sensitization in midget cells. We addressed this question with experiments designed to determine the spatial extent of the sensitization mechanism; we found 1) that sensitization was present for stimuli restricted to the receptive field far surround and 2) that it persisted for stimuli reaching ~ 0.5 mm (2.5 degrees) into the receptive-field surround (Figure 3). A wide-field mechanism in excess of 0.5 mm would be required to exert such effects; these and other data presented in the manuscript clearly indicate that wide-field amacrine cells mediate contrast sensitization in the midget pathway.

A second set of experiments used temporally white noise to directly measure how changes in variance affected midget cells for continuous stimulation at different spatial scales (Figure 5). We found that restricting stimulation primarily to the receptive-field center produced weak contrast adaptation in midget ganglion cells. In the same cells, continuous wide-field stimulation of both center and surround produced contrast sensitization. These data indicate that the midget ganglion cell receptive field is comprised of two competing plasticity mechanisms--a center mechanism that weakly adapts and a surround mechanism that prevents adaptation (i.e., produces sensitization).

Based on comments from two of the reviewers, we performed new analyses of the data in the manuscript to determine how sensitization affected a cell's ability to reliably detect changes in contrast. Two separate measures were used in this analysis: the discriminability index (d') and the Jensen-Shannon distance, a

symmetrical version of the Kullback-Leibler distance. This analysis demonstrated that sensitization significantly improved the ability of midget cells to detect small changes in contrast relative to the background (6-25% contrast). This was true across a variety of stimulus conditions (Figures 2, 3, 6, 8).

Below, we have sought to address these and other points raised by the reviewers.

Reviewer #1 (Remarks to the Author):

Specific comments:

1. It's not obvious from Figure 2 B and 2C that the increased firing rate immediately after adaptation reflects an increase in the contrast sensitivity of the midget cell. The adapted curves in Fig 2B and 8B appear to have an upward shift along the Y axis, suggesting an elevated baseline firing. This is evident by the enhanced spiking at zero contrast. In addition, the gain of the midget cell response is reduced after adaptation as measured by the slope of the curves in Figure 2B, consistent with a reduction in contrast sensitivity. Therefore, the higher firing rate of midget cells after adaptation might result from elevated firing at all contrasts (an upward shift) instead of enhanced contrast sensitivity (a leftward shift). How is "contrast sensitivity" defined in this study? Have the authors considered ways to separate the contrast response from the baseline firing at zero contrast?

This is an excellent point and a similar point was also raised by reviewer #2. Based on these comments, we have performed a careful analysis of the spike responses of midget and parasol cells. We used two metrics for determining how different adapting stimuli affected the ability of these cells to detect changes in contrast relative to the background. The first metric is the sensitivity index (d') and the second is the Jensen-Shannon distance, a symmetrical version of the Kullback-Leibler distance. Both of these metrics show strong agreement--the high frequency adapting stimuli used in our study significantly increased the ability of both On and Off type midget cells to detect low-contrast stimuli (6-25%) relative to the unadapted state. The opposite effect was observed in parasol cells. This pattern held for full-field adapting stimuli, surround stimulation (see below), chromatic stimuli, and background motion.

2. The claim about the wide-field amacrine cell in the abstract seems too strong, because the evidence on the cell type(s) that confers sensitization of bipolar cell inputs is rather indirect and needs substantiation. I suggest a more speculative mention of this hypothesis.

Agreed. We apologize for the incorrect use of this terminology. We were trying to make the point that horizontal cell feedback cannot possibly explain the observed sensitization and the amacrine cell providing the presynaptic inhibition must have a wider receptive field than that of the midget bipolar cells. As you know, however,

the midget bipolar cells have tiny receptive fields in the macula, where we performed the majority of our recordings (~20 μm ; 0.1 degrees). Based on this and other comments, we have performed additional experiments to demonstrate that the mechanism(s) mediating the observed contrast sensitization have a spatial extent well in excess of 500 μm (2.5 degrees; see Figure 3). Horizontal cells and wide-field amacrine cells constitute the only known mechanisms in the primate retina that exert their influences over such large spatial scales. Further evidence excluding horizontal cells and implicating wide-field amacrine cells is presented in the text (Figures 6, 7).

3. According to Figure 5c, there seems to be both a leftward shift of excitation and a rightward shift of inhibition. Could inhibitory mechanism also contribute to the sensitization? It will be helpful to include a discussion about this possibility.

We have reorganized our presentation of our whole-cell data to more clearly reflect the likely effects of synaptic excitation and inhibition in contrast sensitization. Most of the recordings for this study were performed in the central retina (in or near the macula) so that our findings could be more directly related to visual perception. The degree of direct synaptic inhibition onto midget ganglion cell dendrites is dramatically diminished in the central retina relative to peripheral regions (Sinha et al., 2017 Cell). We have included the direct inhibitory synaptic input to the example Off midget ganglion cell in Figure 7. The magnitude of synaptic inhibition in this cell was much smaller than that of excitation. In this figure, we have also shown the horizontal shift data for excitation and inhibition in each cell. Horizontal shift were statistically significant in excitatory currents, but not in inhibition. Further, due to known differences in driving force near spike threshold the magnitude of inhibition must be 3-4 larger than that of excitation in order to exert a meaningful effect on spike output. Thus, we feel confident in stating that the vast majority of contrast sensitization observed in the midget cell spike output arises at or prior to the level of glutamate release from midget bipolar cells. Finally, the lack of sensitization to isoluminant stimuli excludes the horizontal cells as the source presynaptic disinhibition, leaving amacrine cell interactions with the midget bipolar terminal the likely culprit.

4. In the text about the different degree of shifts during wide-field and small-field test flashes, the authors mentioned that “this trend held true across midget cells-horizontal shifts were more negative for the small-diameter test flash than for the wide-field test flash in the same cell... (page 7)”. However, in Fig. 2E, there seems to be an On/Off difference: this trend seems robust for On midget cells, but most of the Off midget cells (4 out of 5) do not show the this trend. Please clarify. In addition, to support the above claim, a statistical analysis to directly compare the shifts under small and wide field conditions are necessary. It would be helpful to

increase the n for this figure, though it is a big ask given the challenges of recording from monkey retinas.

We have separately analyzed the sensitivity metrics for On and Off midget cells for this stimulus paradigm. Following the adapting stimulus, the discriminability index and Jensen-Shannon distance significantly both increased relative to the unadapted control in both On and Off midget cells at contrasts < 0.25 ($p < 0.05$; Wilcoxon signed rank test). For this reason, we conclude that sensitization contributes to contrast coding in both On and Off types. This is now specifically discussed in the text.

5. The sensitizing model showed higher encoding accuracy for periods of fixation relative to periods of ballistic eye movements. However, this cannot be an argument to infer that “sensitization could play a particularly important role in vision during periods of fixation following the offset of global motion”. To support this argument, the authors need to show that the improved performance of the sensitization model over adaptation or no-plasticity model (Δr^2) is larger during fixation than that during ballistic eye movements.

We have modified the model to also calculate the accuracy of the linear-nonlinear model for each of the 161 natural movies. The r^2 values for the adapting and sensitizing models are now shown relative to this LN model that lacks either form of plasticity. The sensitizing model significantly outperforms the LN model at reconstructing the contrast trajectory of these movies.

6. The sensitization model reproduces experimental results during grating stimuli in Figure 6. However, RGC receptive field properties may differ significantly during artificial (such as drifting gratings) and natural stimuli, and they may rely on different modes of spatial integration (for example, Turner et al 2018; Turner and Rieke 2016). Therefore, this model may not accurately predict midget cell responses during natural stimuli. It will be helpful to discuss this potential issue in the discussion.

The reviewer makes an excellent point. We have added a section to the Discussion that specifically addresses the potential shortcomings of our modeling approach in the context of natural vision.

Minor comments:

1. Please indicate statistical significance in the figures and list the sample sizes and p values in the figure legends (e.g Fig.1e, Fig.2c and 2e, Fig. 4d, Fig. 5c, Fig. 8c and 8f) so it would be easier for the readers to find it and comprehend the conclusions.

We have added p-values to these figures along with the statistical tests used to compute these values.

2. Page 4, line 87: “At the transition from high to low contrast”, should it be changed to “... to no contrast” since the schematics in Fig 1A indicate there’s no contrast?

Absolutely. Thank you for catching this. We have fixed the text.

3. Please clarify in the Methods how the x-shift values are calculated from two curves (e.g Fig.2A, 2B and 2D).

We have added a description of how horizontal shifts were calculated in the Methods.

4. Line 71: “to determine whether short-term plasticity in the midget pathway depended on ...”: should be “depends on”

This has been fixed.

5. Text line 161 indicates the contrast as ± 0.5 while figure 3 legend indicates the contrast as $\pm 0.5 - 0.5$.

The text has been updated with the correct range of contrasts.

6. Figure 6B is not referenced in the main text.

A description of this portion of the model have been added both to the main text and to the Methods.

Reviewer #2 (Remarks to the Author):

Sensory neurons adapt to the variance of the input by adjusting their gain and threshold. Classically, ganglion cells were reported to adapt when there is a switch from high variance to low variance by progressively increasing their gain and lowering their threshold. In the transient part right after this switch, the neuron is still adapted to the high variance, and this leads to a very low firing rate and probably a loss in information.

More recently, several works have shown that there is another type of behaviour among ganglion cells, i.e. sensitizing: firing rate increases following the switch and gain is higher. This could compensate for the loss of information, but it could also be that sensitizing cells are not just here to complement classically adapting cells.

The present paper aims at contributing to this in a meaningful way: first, by reporting for the first time sensitizing behaviour in the primate retina. Second, by showing that sensitizing is present in midget ganglion cells, while parasol cells are adapting. This is interesting because these two types compute very different features and are not functional counterparts. This goes against the idea of complementing adapting cells.

Since this is the first report of sensitizing ganglion cells in primate, I think this paper is of significance, especially with recent studies now speculating about the perceptual consequences of sensitization.

However, I have several major concerns that need to be addressed to make this study really convincing:

1) first, it is not clear that the behaviour they found is similar to what was found in the salamander retina. As a consequence, it is not clear if we are really talking about the same thing or not. In particular, the authors report here a small decrease in gain and a change in threshold. Kastner and Baccus, in their seminal paper about contrast sensitization, report an increase in gain. This is an important difference and it is not clear if the analogy still holds. See also my comments below about other differences.

This is a point of confusion for us as well. The original study indicated the slope of the input-output nonlinearity increased for sensitizing cells, as the reviewer points out. However, the second study indicated that the slope (gain) was decreased in sensitizing cells. This is text from the latter work on this point:

“Finally, even though sensitization decreases the threshold during *Learly*, it also decreased the slope in the spiking nonlinearity, as measured from extracellular recordings (Figure S3C). This indicates that sensitization differs from changes in sensitivity due to adaptation, where the slope increases when the threshold decreases (Kastner and Baccus, 2013).”

For sake of clarity, we have removed data interpretation based on horizontal shifts and changes in slope of the input-output nonlinearity. Based on the suggestion of this reviewer below and reviewer #1, we have instead relied on information-theoretic sensitivity metrics in drawing conclusions about effects on neural coding. These metrics make it clear that adapting stimuli evoked opposing effects in midget and parasol cells, increasing sensitivity in midget cells and decreasing sensitivity in parasol cells.

The only exception is that we use horizontal shifts as a descriptive measure of the effects observed in the excitatory and inhibitory synaptic inputs, as obtaining the number of repeats necessary for an information-theoretic analysis is untenable for whole-cell recordings from central midget cells.

Also, the stimuli that are used here are different from the ones in this previous study and this difference should at least be discussed.

2) second, the title and some claims of the abstract are a bit misleading, since they suggest that the sensitization “improves encoding”. This means that the mutual information between the stimulus and the response is higher than if there was no sensitization, but this is never shown in the data. The only place where this issue is tackled is in the model, but it is hard to make a convincing case using only a model whose agreement with data is only qualitative (it is never used to predict ganglion cell responses directly), and noise is not modeled carefully. If the authors want to make this claim, estimating mutual information on the data is necessary (and possible).

Based on these comments and those of reviewer #1, we have performed a careful analysis of the spike responses of midget and parasol cells. We used two metrics for determining how different adapting stimuli affected the ability of these cells to detect changes in contrast relative to the background. The first metric is the sensitivity index (d') and the second is the Jensen-Shannon distance, a symmetrical version of the Kullback-Leibler distance. Both of these metrics show strong agreement--the high frequency adapting stimuli used in our study significantly increased the ability of both On and Off type midget cells to detect low-contrast stimuli (6-25%) relative to the unadapted state. The opposite effect was observed in parasol cells. This pattern held for full-field adapting stimuli, surround stimulation (see below), chromatic stimuli, and background motion.

3) third, the number of cells recorded is low, especially since there is a significant variability in the reported population data (fig2 and even more fig 4 and 5). Also, it seems that ON and OFF cells should be separated in the analysis. It could be that just one of them is sensitizing, there is no good justification to pool them together.

We have addressed both of these concerns in the text. First, we have added additional cells to the study and these cells bolster our original conclusions. Second, we separately analyzed the effects of the adapting stimuli in On and Off type midget cells and found a significant increase in d' and Jensen-Shannon distance in following the adapting stimuli both On and Off midget cells. Our data clearly indicate the presence of sensitization mechanisms for both types. This is now addressed directly in the text.

I have a list of additional concerns which should also be addressed.

-fig1-2, experiment with grating.

Similar to Kastner and Baccus averaging across multiple noise instantiations, it would be better to average firing rate over different stimuli where the initial and final phase of the grating is picked at random. This would also exclude that the transient increase in firing rate is not just a slow ON or OFF response to the last grating presented, or anything phase-dependent.

We modified the grating stimulus and our other adapting stimuli to randomize the phase on each trial and performed the experiments on additional cells. We did not observe either a qualitative or quantitative differences in sensitization between cells in which the phases were randomized and those in which they were static. The text now contains statistical comparisons between these conditions.

-fig 2: color labeling is strange, why not a different color for each contrast level ?

We apologize for the color scheme used in the original manuscript. We have experimented with other color schemes and now present one in which each contrast is color coded and which shows high chromatic contrast. We hope this is to the reviewer's liking. We are also open to other suggested color schemes; specific suggestions would be helpful.

It is not clear how gain is estimated here - methods should be more explicit about this: maybe they fit a sigmoid curve and report the change in parameters? The way it is reported, x-shift, gain etc, is a bit confusing and might be difficult to follow for the naive reader.

We have added details about how we define gain and how the fitting was performed both to the main text and the Methods.

-fig 3: there is some possible confusion in the use of the terms wide field and narrow field, and this confusion is present in the discussion too. My understanding is that the best way to have sensitization is a wide field noise followed by a local probe stimulus, is it correct ? Please clarify this.

Absolutely. We apologize that this was not made clear in the original text. Yes, sensitization was strongest when a small spot was presented following full-field stimulation or texture motion. We have clarified this in the text.

-One can also regret that there is no study of how the sensitization depends on the stimulus location, similar to Kastner and Baccus 2013. Maybe it remains adaptive when the probe stimulus is in the surround, and this explains why the sensitization is weaker in the surround ? Or is it weaker just because of surround suppression ? I don't think a full study on this can be expected, but at least some discussion would be welcomed.

This is an excellent point that we have tried to address directly with additional experiments. We modified our full-field adaptation protocol to adapt only the surround and then we probed the sensitivity of the surround with a series of flashes (Figure 3). This protocol produced sensitization in midget ganglion cells and weak adaptation in parasol ganglion cells. As with stimuli presented over the receptive field center (Figure 2), this surround stimulus paradigm produced a

significant increase in sensitivity for low-contrast stimuli (contrast, ≤ 0.25 ; Figure 3d). We further tested the spatial extent of the surround mechanism in midget cells and found that it extends over diameters well in excess of 500 μm (2.5 degrees; see Figure 3). This is at least 250 times the width of the midget cell receptive-field center at comparable eccentricities. These and other data indicate that sensitization arises from wide-field amacrine cells (see Figure 6, 7).

-fig 3: what is the window over which firing rate is estimated? Is delay defined by the starting point of this window? When is it significantly different from 0? More description of the methods used would be welcomed.

Also, error bars are large and n quite small (see above).

Axis legend: I guess log delay means delay with a log scale? A bit of a misnomer, just "delay" would be better.

Definitely. We thank the reviewer for pointing this out. This has been fixed (see Figure 4).

-fig5D: there should be population analysis at least in the corresponding text.

The increase in excitatory charge during the zero-contrast condition following the adapting stimulus was statistically significant. The values and statistics for the adapted and unadapted conditions are now presented in the text.

-Fig 5: please show the inhibitory counterpart fully with an example of raw data, not just 5C.

The inhibitory currents are now included in the figure (new Figure 7c).

-fig 6: the model predicts adaptation of the amacrine output: do you see it in inhibitory current recordings?

We did not see consistent evidence of adaptation in direct inhibitory synaptic input to midget ganglion cells. In our working model of contrast sensitization in the midget pathway, an amacrine cell with a moderate to large receptive field provides inhibition at the midget bipolar terminal, but not at the level of the midget ganglion cell dendrites. This amacrine cell adapts during high-variance stimulation causing disinhibition at the bipolar terminal when stimulus variance is reduced. We base this model on several key observations. 1) The experiments with isoluminant stimuli exclude the horizontal cells as the source of sensitization (new Figure 6). They also exclude the amacrine cell providing canonical feedforward inhibition to midget cell dendrites---isoluminant stimuli strongly modulate midget bipolar cells in the central retina and the direct inhibition to midget ganglion cell dendrites is directly yoked to the activity of the midget bipolar cell

(Crook et al., 2011). Despite this, an isoluminant adapting stimulus failed to elicit sensitization (Figure 6). 2) Our whole-cell recordings show sensitization in the excitatory synaptic inputs from midget bipolar cells to midget ganglion cells, but not in direct synaptic inhibition (Figure 7). This pattern indicates a presynaptic source for contrast sensitization. Further, these excitatory current recordings show evidence of presynaptic disinhibition---at the offset of the adapting stimulus we observed a bolus of excitatory current onto the midget cell dendrites (Figure 7e).

-fig7: I don't think the modeling study is very insightful as it stands. The purpose of this kind of modeling should rather be to understand what make adapting cells worse at encoding the stimulus, but this study provides no insight about this.

We tested model performance at reconstructing the contrast trajectory in each of the 161 movies, as before. Again, the sensitizing model significantly outperformed the adapting model at this task. We also included the classical linear-nonlinear (LN) model (based on a comment from reviewer #1) in this analysis and found that the sensitizing model outperformed the LN model as well.

Note also that Kastner and Baccus reported a saturation of of the non-linearity in high variance stimuli for sensitizing cells in salamander. For this reason, adapting cells encoded better the stimulus during episode of high variance. Anything similar here ? Answering this question would be a great way to exclude (or not) the hypothesis of complementary pathways.

Thank you for this excellent suggestion. We have performed new experiments in which we presented high-contrast Gaussian noise (s.d., 0.3) to midget and parasol cells. We recovered the nonlinearities for these cells using the classical linear-nonlinear model analysis. The degree of saturation was then quantified by comparing the slopes for the high and low variance regions of the nonlinearity. We found that 1) midget cells lacked significant saturation during high variance stimulation and 2) the saturation in midget (sensitizing) and parasol cells (adapting) was similar. This, along with the clear functional distinction of midget cells relative to other primate ganglion cell types, indicates that sensitization plays a distinct role in the midget pathway relative to its role in other vertebrate species.

Maybe the model can also be used to explore the range of statistics where sensitizing cells encode better. Here we are left with the impression that they also do a better job at this than adapting cells, even if the discussion says that adapting cells are better "to infer when salient properties of the environment change". If you are always better at encoding the stimulus is not clear how you will be worse at detecting motion. Please clarify.

Based on the comment presented here, we sought insight into the stimulus conditions in which neural adaptation would improve encoding. Previous work

supported a role for adaptation in determining when salient stimulus features changed (Fairhall et al., 2001; Mlynarski and Hermundstad, 2018; Wark et al., 2009). We tested this hypothesis directly by fitting our models to the change in contrast as a function of time. Indeed, the adaptation model outperformed both the sensitizing and linear-nonlinear models at reproducing the change in contrast (Figure 9). In fact, the adaptation model performed significantly better at encoding the change in contrast than at encoding the contrast itself---model correlation improved by 149% for the linear decoding paradigm and 27% for the quadratic decoding paradigm (Figure 9).

-fig 8: no increase in activity similar to figure 2. Why ? The model would predict such an increase. Actually, if anything, there might be a weak increase for parasol cells, but maybe not significant.

We apologize if the figure and the accompanying text of the original manuscript were unclear. We are not certain to which aspect of this figure the reviewer is referring in this comment. Background motion caused parasol cells to adapt, decreasing their sensitivity to low contrast stimuli (new Figure 10a-c) while midget cells increased their sensitivity following background motion (Figure 10d-f). These increases in sensitivity were significant at low contrast (≤ 0.5) just as they were for the wide-field temporal adapting stimulus (Figure 2).

Eccentricity of the recorded cells should also be reported, as it could affect the results and their interpretation significantly, especially regarding to connection to psychophysics.

This is an excellent point. We specifically focused on recording from midget cells in macula so that our findings could be related to psychophysical work. The eccentricity of our cells is evident their responsiveness to isoluminant stimuli (Figure 6) as more peripheral midget cells lose sensitivity to purely chromatic stimuli (Wool et al., 2018 *J Neurosci*) and also in the small magnitude of direct synaptic inhibition (Sinha et al., 2017 *Cell*). We have added more detailed information about the foveal eccentricity of recorded cells to the main text and Methods.

Reviewer #3 (Remarks to the Author):

This paper demonstrates and explains the mechanisms underlying sensitization of visual responses in midget ganglion cells of the primate retina. The authors show that a wide-field, low spatial frequency stimulus reduces subsequent responses in parasol cells (i.e., adaptation) but enhances responses in midget ganglion cells (facilitation). Facilitation (or “sensitization”) lasts for 100’s of ms and is greatest when the test stimulus is small and centered in the middle of the cell’s receptive field. The authors go on to show very nicely that the circuit is sensitized following

achromatic stimulation, arguing against a significant role for horizontal cell-cone feedback mechanisms. Sensitization is present in the excitatory synaptic inputs to midget ganglion cells, indicating that it is not due to postsynaptic integration of excitatory and inhibitory inputs. Instead, the results point to lateral feedback inhibition from amacrine cells onto presynaptic bipolar cell terminals as

the primary mechanism. Final experiments show that sensitization occurs in response to background motion similar to a saccadic eye movement, adding to the idea in the field that inhibitory circuitry helps the retina account for self-motion. A mathematical model nicely suggests that sensitizing circuitry likely does a better job reconstructing natural stimuli than adapting circuitry.

This is a very nice paper. The experiments are carefully done, nicely analyzed and (for the most part) clearly presented. All of my comments, save one, are simple suggestions to slightly enhance the clarity of the presentation.

There is a lot of work in this paper, and I hesitate to ask for another experiment, so I hope the authors (and editors) will take this as a suggestion rather than a requirement. The authors show that lateral inhibition underlies sensitization. It would be interesting to examine the spatial scale of this inhibition, i.e., how does sensitization depend on the diameter of annular surround stimulation? How does this spatial scale compare to that of presynaptic inhibition as detected in EPSCs recorded in the midget ganglion cells? As the field anticipates a primate retinal connectome, this information would be invaluable for subsequently identifying the amacrine cell(s) mediating the sensitizing mechanism.

This is an excellent question that was also brought up by reviewer #2. We have attempted to address this question by restricting the adapting and probe stimuli to the receptive-field surround (see Figure 3). This stimulus paradigm produced significant increases in sensitivity at low contrast ($\leq 25\%$). The mask diameter of this stimulus was 80-160 μm , which is much larger than the receptive field center size of midgets that we recorded ($\leq 40 \mu\text{m}$).

To carefully measure the spatial extent of this surround effect, we measured surround sensitization with mask diameters of 160-640 μm in the same cell. We found significant increases in sensitivity for mask diameters $\leq 480 \mu\text{m}$. Based on the assumption of an approximately Gaussian weighting of visual signals, the wide-field amacrine cell mediating contrast sensitization in the midget pathway must have a dendritic extent well in excess of 500 μm (> 2.5 degrees) in order to exert the observed effects. This should narrow the list of candidate cell types for future studies. [redacted]

Minor comments:

In the Results, you describe the midget data in figure 1 prior to the parasol/thorny data, yet the figure is organized in the opposite order. It seems more logical to place the midget data in panels A and B.

This is an excellent point. We have changed the presentation order for these data in Figure 1. The accompanying text and figure legend have been updated to reflect this change.

Page 9: The description of the results in Figure 4 is well-written, but it would be helpful to the reader if the authors would make more specific references in the text to relevant panels in the figure.

We thank the reviewer for pointing out this oversight. The description of Figure 4 (now Figure 6) in the original text was clearly inadequate. We have updated the text with references to specific figure panels to improve the clarity of this section of text.

Line 314: "Differencing": I was surprised to find that this is actually a word, at least in the context of heraldry, but I think "subtracting" is the better choice.

Well put. We have changed the text according to the reviewer's suggestion.

Reviewers' Comments:

Reviewer #1:

Remarks to the Author:

The authors have addressed all my concerns. This is a very nice study.

Reviewer #2:

Remarks to the Author:

The authors have largely addressed my comments (sometimes they went much deeper and better than I expected). This is a really nice paper. I have a couple of suggestions to improve readability but if the authors don't think it will help they should not implement them.

Fig 5b, right panel: maybe shift the x axis so that 0 is the time of the switch between high and low contrast ?

Put the center and surround as circles in the images describing the stimulus in fig 3 ?

Use of the term 'receptive field': in some cases it might not be clear to the reader if they mean 'receptive field center' or center+surround. A possible alternative would be to use RF center when appropriate, and center and surround otherwise. But I leave this up to the authors.

Reviewer #3:

Remarks to the Author:

The authors have responded conscientiously to the previous reviews, and I have no other comments or concerns. This is a very nice paper.

Reviewers' Comments:

REVIEWERS' COMMENTS: Reviewer #1 (Remarks to the Author):

The authors have addressed all my concerns. This is a very nice study.

Reviewer #2 (Remarks to the Author):

The authors have largely addressed my comments (sometimes they went much deeper and better than I expected). This is a really nice paper. I have a couple of suggestions to improve readability but if the authors don't think it will help they should not implement them.

Fig 5b, right panel: maybe shift the x axis so that 0 is the time of the switch between high and low contrast ?

This an excellent suggestion. We modified the figure accordingly.

Put the center and surround as circles in the images describing the stimulus in fig 3 ?

We have updated this figure to explicitly indicate the center and surround regions and more clearly describe the stimulus paradigm.

Use of the term 'receptive field': in some cases it might not be clear to the reader if they mean 'receptive field center' or center+surround. A possible alternative would be to use RF center when appropriate, and center and surround otherwise. But I leave this up to the authors.

We thank the reviewer for this suggestion. We have modified the text in several places to more carefully describe whether the stimuli used engaged the center or surround regions of the receptive field.

Reviewer #3 (Remarks to the Author):

The authors have responded conscientiously to the previous reviews, and I have no other comments or concerns. This is a very nice paper.